# Hyperthermophilic methanogenic archaea act as high-pressure CH$_4$ cell factories

Lisa-Maria Mauerhofer[1], Sara Zwirtmayr[2], Patricia Pappenreiter[2], Sébastien Bernacchi [3], Arne H. Seifert[3], Barbara Reischl[1,3], Tilman Schmider[1], Ruth-Sophie Taubner [1,2], Christian Paulik [2] & Simon K.-M. R. Rittmann [1✉]

Bioprocesses converting carbon dioxide with molecular hydrogen to methane (CH$_4$) are currently being developed to enable a transition to a renewable energy production system. In this study, we present a comprehensive physiological and biotechnological examination of 80 methanogenic archaea (methanogens) quantifying growth and CH$_4$ production kinetics at hyperbaric pressures up to 50 bar with regard to media, macro-, and micro-nutrient supply, specific genomic features, and cell envelope architecture. Our analysis aimed to systematically prioritize high-pressure and high-performance methanogens. We found that the hyperthermophilic methanococci *Methanotorris igneus* and *Methanocaldococcccus jannaschii* are high-pressure CH$_4$ cell factories. Furthermore, our analysis revealed that high-performance methanogens are covered with an S-layer, and that they harbour the amino acid motif Tyr$^{\alpha444}$ Gly$^{\alpha445}$ Tyr$^{\alpha446}$ in the alpha subunit of the methyl-coenzyme M reductase. Thus, high-pressure biological CH$_4$ production in pure culture could provide a purposeful route for the transition to a carbon-neutral bioenergy sector.

[1] Archaea Physiology & Biotechnology Group, Department Functional and Evolutionary Ecology, Universität Wien, Wien, Austria. [2] Institute for Chemical Technology of Organic Materials, Johannes Kepler Universität Linz, Linz, Austria. [3] Krajete GmbH, Linz, Austria. ✉email: simon.rittmann@univie.ac.at

Methane ($CH_4$) is an energy carrier of worldwide importance. It can be produced through biogenic, thermogenic, and pyrogenic processes[1]. Most biogenic $CH_4$ is emitted by methanogenic archaea (methanogens)[2], with minor amounts originating from cyanobacteria[3] and marine microorganisms[4]. Methanogens are a phylogenetically diverse group of microorganisms, which can be found in various anoxic environments[5]. Among other substrates, methanogens convert short chain organic acids and one-carbon compounds to $CH_4$ through their energy and carbon metabolism[2,5,6]. Their metabolic capability is important for anaerobic organic matter degradation in environments with low concentrations of sulfate, nitrate, manganese, or iron[5]. Moreover, methanogens are of biotechnological relevance due to their ability to produce isoprenoid-containing lipids[7,8] or polyphosphate[9], and were recently described to excrete proteinogenic amino acids[8]. Methanogens are central to biofuels production, as they can be employed as autobiocatalysts for carbon dioxide ($CO_2$) and molecular hydrogen ($H_2$) conversion in the $CO_2$-based biological $CH_4$ production ($CO_2$-BMP) process.

The $CO_2$-BMP process can be employed in multiple applications such as biogas upgrading, power-to-gas applications, decentralized energy production, and for the conversion of $H_2/CO_2$ of process flue gasses in waste to value concepts from, e.g., ethanol, petroleum, steel, and chemical industries[10]. There are two main approaches for $CO_2$-BMP[11]: ex situ biomethanation using pure cultures[12,13] or enriched mixed cultures[14–16], and in situ biomethanation[17,18]. In situ biomethanation is examined for upgrading the $CH_4$ content of biogas by adding $H_2$ to anaerobic digesters. Ex situ pure culture biomethanation exhibits high volumetric $CH_4$ productivity and offers a straightforward bioprocess control by utilizing biochemically and biotechnologically well-characterized microorganisms in pure culture[12]. Among the most studied organisms in this regard is *Methanothermobacter marburgensis*[12,19], exhibiting several advantageous traits such as flexibility with regard to substrate gas impurities[10] and high $CH_4$ productivity[20]. In addition, *M. marburgensis* can be used for $CO_2$-BMP when short-term transitions in the order of minutes are demanded between stand-by to full load biomethanation. Furthermore, downtime periods above 500 h did not reduce $CH_4$ productivity after a process restart[21].

Compared to $CO_2$-BMP, chemical methanation or the "Sabatier reaction" should not be operated intermittently due to various catalytic constraints[22] and the fast bulk-like oxidation of the nickel catalyst in the $CO_2$ atmosphere[23]. Furthermore, activity loss of the chemical catalyst after a certain lifespan necessitates the exchange of the catalyst and the carrier material leading to periodic downtimes in production. Thus, applying methanogens, which are autobiocatalysts, offers numerous advantages compared to a chemically catalyzed $CO_2$ methanation. The lower power demand and the stable selectivity observed in $CO_2$-BMP compared to chemical methanation[22] strongly suggest that $CO_2$-BMP is a viable biotechnological alternative to chemical methanation. However, the autobiocatalytic characteristics of methanogens require further investigation.

The $CO_2$-BMP bioprocess can be operated as a gas transfer limited process[12] when a proper feeding strategy is applied[24]. In this case, the kinetic limiting step is the mass transfer of $H_2$ to the liquid phase. In biochemical engineering, gas to liquid mass transfer can be enhanced by several technical measures[20]. Besides reactor geometry and agitation, which influence the specific mass transfer coefficient ($k_La$), pressure increases the solubility of $H_2$ in the liquid phase. The influence of pressure on substrate uptake, growth, and production kinetics of methanogens is therefore an important parameter in $CO_2$-BMP. Some experiments with *Methanocaldococcus jannaschii* have already been performed at high pressure in order to investigate transcription profiles[25] or

growth and $CH_4$ production[26]. The effect of pressure on $CH_4$ production has also been examined in bioreactors[20,27], while media for cultivation of methanogens have been developed and their growth assessed[28–31]. However, a systematic biotechnological survey with regard to nutritional demands of methanogens across different temperature regimes in the same cultivation conditions and at different pressure levels has not yet been the focus of any study.

On the way to develop a high-pressure pure culture $CH_4$ production bioprocess, we systematically and quantitatively investigated the productivity of methanogens at pressures up to 50 bar. Growth, conversion, and $CH_4$ productivity were first examined in order to identify cell factories with the highest $CH_4$ productivity among 80 methanogens, in a range of different media (in terms of composition and medium amendments) and in conditions ranging from psychrophilic to hyperthermophilic. Secondly, the 14 prioritized fastest growing and with the highest productivity methanogens were investigated using a high frequency gassing (HFG) experiment and by using 10 bar $H_2/CO_2$ to $CH_4$ conversion experiments. Among these 14 methanogens, four strains were chosen for the third step, consisting of 50 bar $H_2/CO_2$ to $CH_4$ conversion experiments. Finally, we analyzed these results in the context of their natural habitat, temperature optima, specific genomic features, and their cell envelope architecture.

## Results

**High-throughput screening revealed high-performance methanogens**. In order to investigate essential macro- and micronutrient growth medium amendments of methanogens in the context of their physiology and $CH_4$ productivity in a systematic and quantitative physiological approach, a multivariate high-throughput screening of 80 methanogens on various media and medium amendments was performed (Fig. 1 and Supplementary Figs. S1 and S2). This multivariate high-throughput screening was conducted on 22 complex and defined media in order to characterize methanogens from psychrophilic, mesophilic, thermophilic, and hyperthermophilic temperature groups with regard to maximum biomass and $CH_4$ production kinetics in a closed batch setting, with initially 2 bar $H_2/CO_2$ (4:1) in the headspace. Maximum optical density ($OD_{max}$), maximum substrate conversion ($turnover_{max}$), and maximum volumetric $CH_4$ evolution rate ($MER_{max}$) were selected as experimental output variables. To elucidate if the chosen methanogens showed a homogenous or heterogenous growth pattern which would indicate a balanced or unbalanced biomass increase[32], respectively, the biomass increase rate ("Material and Methods", Eq. (1)) was used for comparing growth kinetics.

All tested psychrophilic methanogens grew to an $OD_{max}$ of below 0.2, showed $turnover_{max}$ lower than 40%, and exhibited a very low $MER_{max}$ of only up to 0.1 mmol $L^{-1}$ $h^{-1}$ on complex or defined media. Mesophilic methanogens grew heterogeneously when cultivated on complex and/or defined media. Biomass growth to an $OD_{max}$ beyond 1.0, a $turnover_{max}$ over 70%, and a $MER_{max}$ higher than 1.0 mmol $L^{-1}$ $h^{-1}$ at cultivation temperatures between 35 and 37 °C on complex media were measured. Growth of moderate thermophilic methanogens between 40 and 45 °C resulted in an $OD_{max}$ between 0.3 and 0.7 on complex media, while on defined media only an $OD_{max}$ of 0.003–0.025 was obtained. Growth of thermophilic methanogens (60–65 °C) resulted in an $OD_{max}$ range from 0.4 to 0.8, a $turnover_{max}$ between 75 and 96% and a $MER_{max}$ beyond 1.0 mmol $L^{-1}$ $h^{-1}$. Hyperthermophilic growth between 80 and 98 °C resulted in an $OD_{max}$ between 0.2 and 0.7, a $turnover_{max}$ from 83 to 97% and a $MER_{max}$ ranging from 1.0 to 4.6 mmol $L^{-1}$ $h^{-1}$ (Fig. 1 and Supplementary Fig. S2). Most methanogens showed a

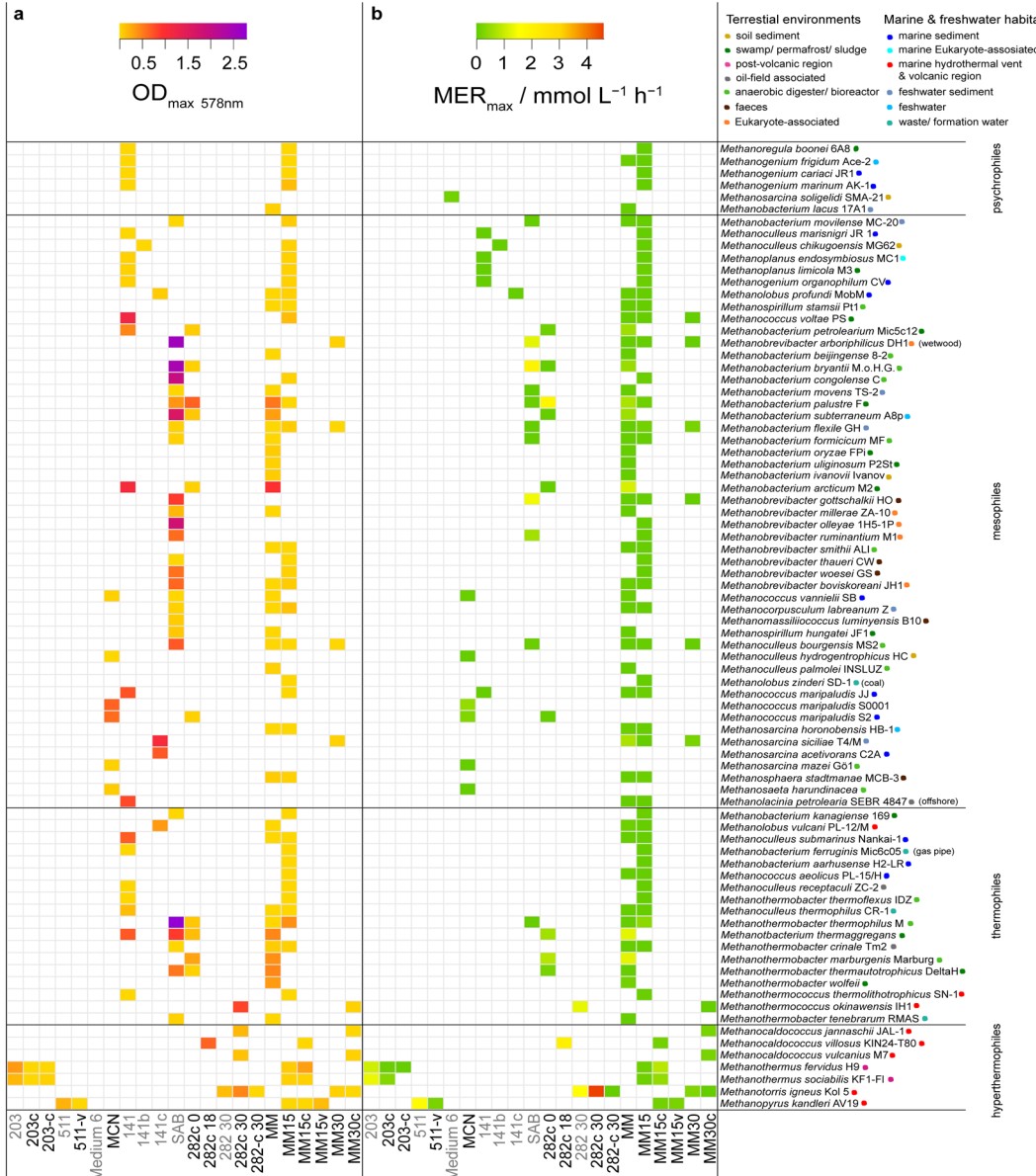

**Fig. 1 Biomass and CH$_4$ production kinetics of the multivariate prescreen of 80 methanogens in defined and complex media.** Experiments were performed in closed batch cultivation systems at 2 bar (120 mL flasks, 50 mL medium). On the *y*-axis, methanogens were arranged as groups according to their temperature optimum in psychrophiles, mesophiles, thermophiles, or hyperthermophiles. Methanogens are listed with ascending strain-specific temperature optimum from top to bottom. Coloured points next to the strain designation on the *y*-axis indicate the isolation site of the tested methanogen (terrestrial habitats: golden brown—soil sediment, dark green—swamp/permafrost/sludge, pink—post-volcanic region, gray—oil-field associated, light green—anaerobic digester/bioreactor, brown—feces, orange—eukaryote-associated; marine and freshwater environments: bright blue—marine sediment, turquoise—marine eukaryote-associated, red—marine hydrothermal vent and volcanic region, gray blue—freshwater sediment, sky blue—freshwater, green blue—waste/formation water). In total, 22 defined and complex media were tested, but not every strain was cultivated on every medium. Defined and complex media are shown on the *x*-axis in black and gray fonts, respectively. For each closed batch cultivation, three biological replicates (in some cases, two biological replicates) plus one negative control were used. **a** The maximum absorption is shown as OD$_{max}$ at 578 nm, and **b** the maximum volumetric CH$_4$ production rate is shown as MER$_{max}$ / mmol L$^{-1}$ h$^{-1}$.

homogenous growth pattern and a biomass increase rate below 10 (Supplementary Fig. S1). Interestingly, mesophilic methanogens grown on SAB (complex medium) showed a biomass increase rate between 10 and 40 (Supplementary Fig. S1).

**Correlating nutritional demands, growth, and CH$_4$ productivity.** In order to correlate nutritional demands to associated growth, substrate conversion, and productivity, a standardized principal component analysis (PCA) and subsequent k-means

cluster analysis was performed. The cluster analysis was performed for OD$_{max}$, turnover$_{max}$, MER$_{max}$, and the combination of those variables together with the concentrations of sulfate and/or sulfur, ammonium, phosphate, and cysteine in the respective media. These data were then linked to strain-specific information such as taxonomy and cultivation temperature. Further, medium-associated parameters were used for interpretation, such as the applied medium with the corresponding trace element solution (TES), and the addition of vitamin solution (VS), cysteine, and yeast/peptone to the medium. The clustering approach then

enabled the grouping of the mentioned variables and parameters into clusters.

High OD values were achieved on complex medium with VS, cysteine, and yeast extract/peptone (Supplementary Data 1, Table S1 and Supplementary Fig. S3). Mesophilic strains *Methanococcus* spp., *Methanobacterium* spp., and thermophilic methanogens belonging to the genera *Methanothermobacter* and *Methanobacterium* grew on defined media without vitamins and cysteine (MM medium, Fig. 1 and Supplementary Data 1, Table S1). At least 50% reduction of $OD_{max}$ was observed with strains that grew best on MM medium when cultivated on 282c 0 medium (rich-TE and cysteine addition), except for *Methanobcterium palustre*. This strain showed similar growth on MM and 282c 0 medium indicated by an $OD_{max}$ of 0.53 and 0.66. *M. palustre* also showed higher $CH_4$ production kinetics on 282c 0 medium compared to MM medium, indicated by a turnover$_{max}$ up to 93% and a 2.6-fold higher $MER_{max}$ of 1.9 mmol $L^{-1} h^{-1}$ in 282c 0 medium (Fig. 1 and Supplementary Data 1, Table S2). Methanogens that require supplements (VS or cysteine) in the media, like Methanopyri or Methanococci, reached $OD_{max}$ values between 0.2 and 0.8.

The highest turnover was achieved by methanogens grown in defined media. Methanococci and Methanobacteria showed a turnover$_{max}$ between 90 and 98% on medium 203 and 282-based media (Supplementary Data 1, Table S3, and Supplementary Fig. S4). A turnover$_{max}$ between 80 and 90% was achieved on media MM, MCN, SAB, and 511. The highest turnover$_{max}$ values (90–98%) were obtained in a medium with 30 times lower phosphate concentration and five times lower ammonium concentration compared to the media used in the turnover$_{max}$ range 80–90% (Supplementary Data 1, Table S3, and Supplementary Fig. S4), which might indicate that phosphate and ammonium concentrations in the medium need optimization.

Methanococci, Methanobacteria, and Methanopyri were found to be highly productive in a closed batch cultivation mode at 2 bar, indicated by a $MER_{max}$ range from 1.1 to 4.6 mmol $L^{-1} h^{-1}$ (Supplementary Data 1, Table S4, and Supplementary Fig. S5). These methanogens grew on 282, MM-based media, 511, SAB, and 203 media. The highest $MER_{max}$ values of 2.1 and 4.6 mmol $L^{-1} h^{-1}$ were achieved by Methanococci on 282-based media without yeast/peptone and the addition of cysteine (cluster 1, Supplementary Data 1, Table S4, and Supplementary Fig. S5). The highest $MER_{max}$ of 4.61 mmol $L^{-1} h^{-1}$ was measured for *Methanotorris igneus*. Methanococci showed the highest turnover$_{max}$ and $MER_{max}$ values (Supplementary Figs. S4–S6, Supplementary Data 1, and Table S5), assuming that growth and $CH_4$ productivity is positively influenced by sulfate, sulfur, and cysteine.

On defined medium without cysteine (MM-based and MCN medium), mesophilic methanogens from the order Methanobacteria, Methanomicrobia, and Methanococci (with the exception of *Methanothermus fervidus*) reached a $MER_{max}$ range of 0.5 to 1.0 mmol $L^{-1} h^{-1}$ (Supplementary Data 1 and Table S4). Strains isolated from hyperthermophilic environments (Fig. 1) like Methanocaldococcaceae and Methanopyraceae required cysteine or vitamins in the medium to exhibit high MERs (282-based and 511 media). They showed growth to an $OD_{max} < 0.03$, and a turnover$_{max}$ and $MER_{max}$ reduction of 90% and 95%, respectively, when cysteine or vitamins were excluded from the media (282-c 30, 511-v, Fig. 1). Attempts to restore the MER of *Methanocaldococcus* spp., *Methanothermococcus* sp., *Methanothermus* sp., and *Methanopyrus* sp. on MM medium at their optimum salt concentration through the addition of cysteine or vitamins after they had been grown in media without these compounds did not recover their $CH_4$ productivity, except for *M. fervidus*. This organism showed an $OD_{max}$ of 0.34 on 203 and MM15c medium. However, the productivity of *M. fervidus* on medium MM15c was

decreased compared to 203 medium, indicated by a $MER_{max}$ of 0.9 and 1.1 mmol $L^{-1} h^{-1}$ (Fig. 1, Supplementary Data 1, and Table S1).

From this comprehensive multivariate, quantitative analysis of growth and $CH_4$ production kinetics, we prioritized *Methanobacterium* spp. and *Methanococcus* sp. (mesophilic), *Methanothermobacter* spp., *Methanobacterium* sp., and *Methanothermococcus* sp. (thermophilic), and *Methanocaldococcus* spp., *Methanothermus* sp., *Methanotorris* sp., and *Methanopyrus* sp. (hyperthermophilic) for the subsequent 10 bar $H_2/CO_2$ conversion experiments. These methanogens were selected due to their ability to grow fast on defined media (cluster 1 and 3 in Supplementary Data 1, Table S2, and Supplementary Fig. S3), and their successful reactivation after dormancy (Supplementary Table S1). Prioritized high-performance methanogens showed a turnover$_{max} > 70\%$ (cluster 1, 2, and 4 in Supplementary Data 1, Table S3, and Supplementary Fig. S4) and 90% of the strains exhibited a $MER_{max} > 1$ mmol $L^{-1} h^{-1}$ (cluster 1, 3, and 4 in Supplementary Data 1, Table S4, and Supplementary Fig. S5).

**Identification of high-performance methanogens**. The multivariate quantitative comparative investigation resulted in the prioritization of 14 fast converting and/or fast growing autotrophic hydrogenotrophic methanogens (Fig. 2). Prioritized methanogens belong to Class I methanogens[26]. In a closed batch cultivation system, gas-utilizing methanogens experience extreme gas-limiting conditions, as the substrate (gas in the headspace of the cultivation vessel) is converted at a decreasing rate[27]. To reduce the effect of gas-limiting conditions for fast converting methanogens during closed batch cultivation, HFG experiments were conducted. During HFG experiments the headspace of the serum bottle was replenished with $H_2/CO_2$ before turnover$_{max}$ was reached. Thus, we found that the mesophilic methanogen *Methanococcus maripaludis* and the hyperthermophilic methanogens *M. jannaschii* and *Methanocaldococcus vulcanius* showed a turnover rate $>5 h^{-1}$ (Supplementary Fig. S7). Furthermore, HFG experiments enabled a quantitative and comparative analysis of MERs and biomass increase rates among prioritized methanogens (Fig. 2).

The biomass increase rates of *M. palustre*, *Methanothermobacter thermophilus*, *Methanobacterium thermaggregans*, and *M. vulcanius* varied, indicating a heterogeneous growth pattern (Fig. 2a). Based on the position of the median in the boxplot figures (Fig. 2a), we could observe that *M. thermophilus* and *M. thermaggregans* have a longer lag phase compared to the other tested methanogens. All other strains showed little variation regarding the biomass increase rates, indicating that these strains had a homogenous growth pattern under the tested conditions (Fig. 2a). *M. vulcanius* and *M. jannaschii* showed a MER median value $>5$ mmol $L^{-1} h^{-1}$, which is 1.5-fold the amount compared to *Methanothermococcus okinawensis* and twice the amount of *M. igneus* as well as approximately four times higher compared to the other tested methanogens (Fig. 2b).

**Methanogens with a surface protein layer are $CH_4$ cell factories**. The highest MERs were observed for hyperthermophilic methanogens that grow within a temperature range of 80–85 °C on defined medium 282c 30 and known to possess a surface protein layer (S-layer)[33,34]. These S-layer proteins harbor the InterPro domains IPR022651 S_layer_C, IPR006454 S_layer_MJ, and IPR022650 S_layer_N (Supplementary Data 2). Furthermore, *M. fervidus* is covered with an S-layer protein (slgA in a p6 lattice pattern)[35]. The slgA consists of six InterPro features, IPR006633 Carb-bd_sugar_hydrolysis-dom, IPR007742 NosD_-dom, IPR022441 Para_beta_helix_rpt-2, IPR006626 PbH1,

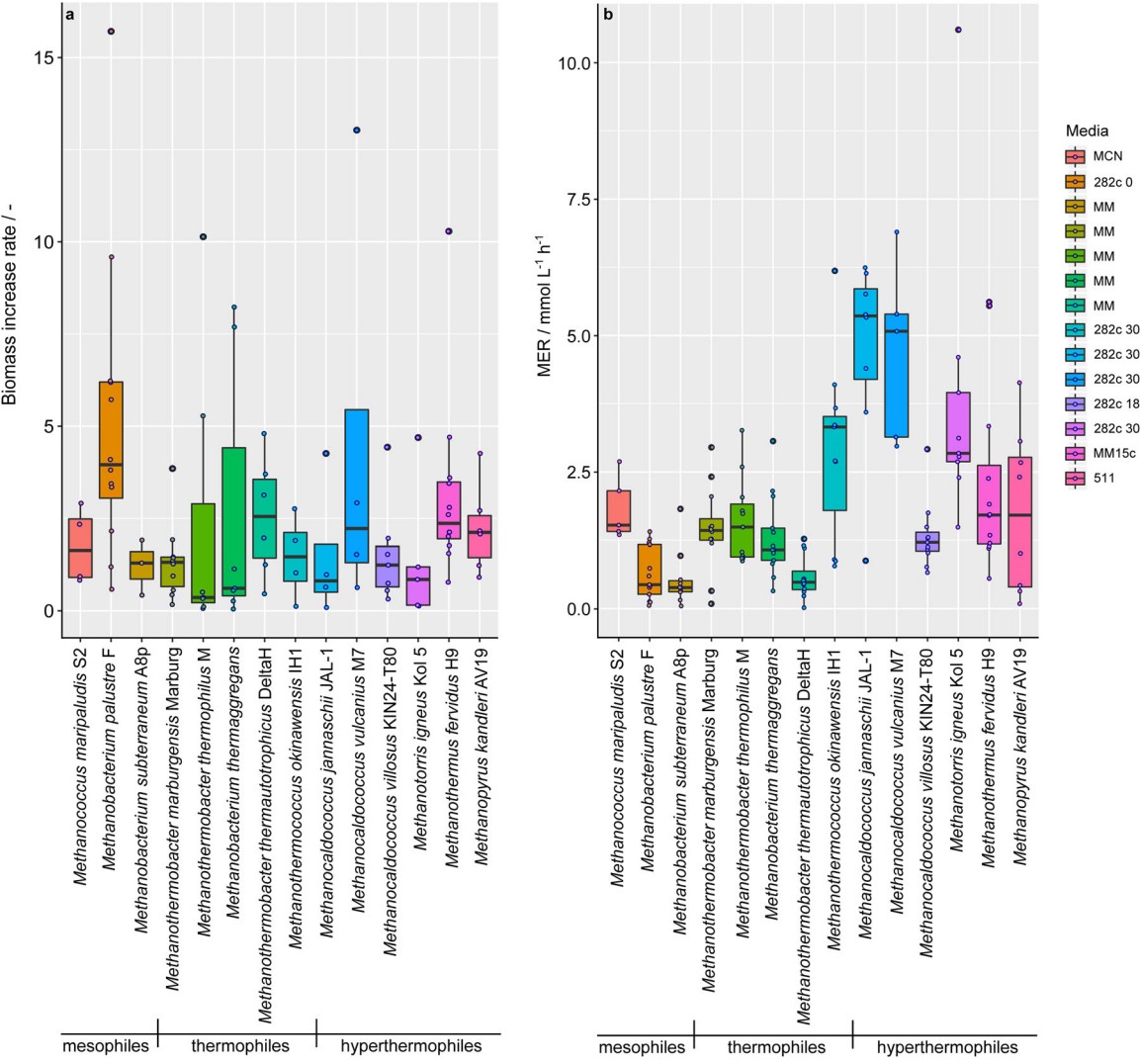

**Fig. 2 Results of high frequency gassing (HFG) experiments of prioritized methanogens at 2 bar in defined medium.** All experiments were performed in quadruplicates including a negative control. **a** The biomass increase rate/- and **b** The MER / mmol L$^{-1}$ h$^{-1}$. Boxplots are used for data visualization. In both subfigures, three temperature blocks are distinguished and highlighted from left to right. Left block: mesophilic methanogens grown at 37 °C (*Methanococcus maripaludis* S2, *Methanobacterium palustre* F, *Methanobacterium subterraneum* A8p); middle block: five thermophilic strains grown at 65 °C (*Methanothermobacter marburgensis* Marburg, *Methanothermobacter thermophilus* M, *Methanobacterium thermaggregans*, *Methanothermobacter thermautotrophicus* DeltaH, *Methanothermococcus okinawensis* IH1); right block: six hyperthermophilic methanogens (*Methanocaldococcus jannaschii* JAL-1 (80 °C), *Methanocaldococcus vulcanius* M7 (80 °C), *Methanocaldococcus villosus* KIN24-T80 (80 °C), *Methanotorris igneus* Kol 5 (85 °C), *Methanothermus fervidus* H9 (80 °C), *Methanopyrus kandleri* AV19 (98 °C)).

IPR012334 Pectin_lyas_fold, IPR011050 Pectin_lyase_fold/virulence as well as the Pfam motif PF05048 NosD. The same motifs can be found in the genomes of *M. marburgensis* and *M. thermautotrophicus* (Supplementary Data 2). This might be an indication for the presence of S-layers on the cell envelop of *M. marburgensis* and *M. thermautotrophicus*. However, up to now S-layers were never described for these organisms, and the function of these homologous proteins would therefore require characterization. Furthermore, the IPR032812, IPR013783 Ig-like_fold, and IPR032812 SbsA_Ig features can be found in the genome of *M. marburgensis*, with the later Ig-like domain present in the S-layer protein SbsA. While *M. thermautotrophicus* does not harbor an IPR032812 SbsA_Ig feature, it encodes an IPR013783 Ig-like_fold feature, a motif which was also detected in *M. vulcanius* and *M. villosus*. *M. kandleri* was shown earlier to possess a S-layer[36], although no S-layer related motifs or domains could be found in our in silico analysis (Supplementary Data 2), but the

IPR011330 Glyco_hydro/deAcase_b/a-brl and IPR002509 NODB_dom features were detected.

**High-performance methanogens harbor a specific MCRα amino acid motif.** A subsequent bioinformatic examination of the key enzyme for methanogenesis, methyl-coenzyme M reductase (MCR), and especially the alpha subunit of the MCR (MCRα) revealed that all highly productive prioritized methanogens harbor the Tyr$^{\alpha444}$ Gly$^{\alpha445}$ Tyr$^{\alpha446}$ amino acid motif and belong to the Class I methanogens (Supplementary Fig. S8). Borrel et al. showed that Tyr$^{\alpha444}$ is substituted to phenylalanine in some *Methanocella* spp., *Methanoregula* spp., *Methanocorpusculum* spp., and *Methanosarcina* spp.[37]. We found that Tyr$^{\alpha444}$ to Phe$^{\alpha444}$ is specific to Class II methanogens, except to *Methanimicrococcus blatticola* and *Methanolinea tarda* (Supplementary Fig. S8). In Class I methanogens, Tyr$^{\alpha444}$ anchors the coenzyme

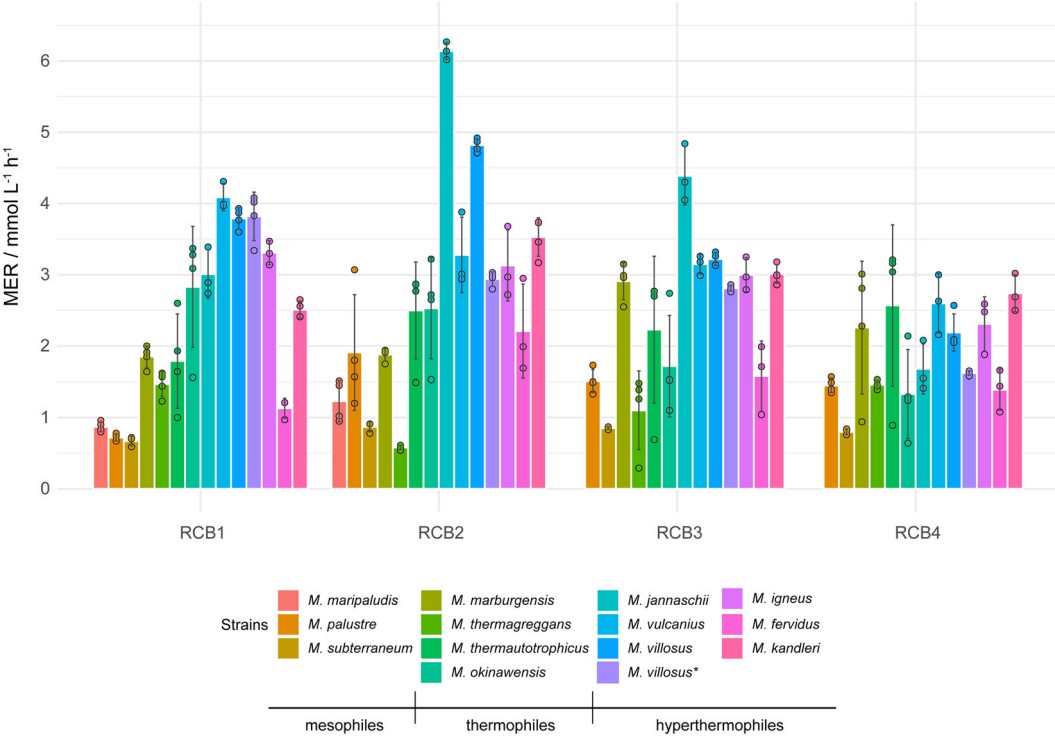

**Fig. 3 Results of high-pressure RCB cultivations of prioritized methanogens in the simultaneous bioreactor system (SBRS) at 10 bar.** All experiments were performed in quadruplicates in the SBRS system at a gassing ratio of $H_2/CO_2$ (4:1)[35]. Mean and standard deviation are shown. The principle of the cultivation was to repressurize each of the bioreactors to 10 bar after full headspace gas conversion. RCB1, RCB2, RCB3, and RCB4 indicate results from individual and successive closed batch headspace gas conversions. MER / mmol $L^{-1} h^{-1}$ is shown. The left block indicates mesophilic methanogens grown at 37 °C (*Methanococcus maripaludis* S2, *Methanobacterium palustre* F, *Methanobacterium subterraneum* A8p); the middle block shows the thermophilic methanogens grown at 65 °C (*Methanothermobacter marburgensis* Marburg, *Methanobacterium thermaggregans*, *Methanothermobacter thermautotrophicus* DeltaH, *Methanothermococcus okinawensis* IH1); and the right block shows hyperthermophilic methanogens (*Methanocaldococcus jannaschii* JAL-1 (80 °C), *Methanocaldococcus vulcanius* M7 (80 °C), *Methanocaldococcus villosus* KIN24-T80 (80 °C), *Methanocaldococcus villosus* KIN24-T80*-grown on 282c 18_E medium (80 °C), *Methanotorris igneus* Kol 5 (85 °C), *Methanothermus fervidus* H9 (80 °C), *Methanopyrus kandleri* AV19 (98 °C)).

M together with two other amino acid residues in the catalytic center of the MCR[38]. Additionally, Tyr$^{\alpha446}$ is predominately exchanged to phenylalanine in Methanosarcinaceae, *Methanosarcina* spp.[37], *Methanohalophilus* spp., *Methanohalobium* sp., *Methanococcoides* spp., *Methanolobus* sp., and *Methanomethylovorans* sp.. Moreover, we found an amino acid exchange from Tyr$^{\alpha446}$ to Phe$^{\alpha446}$ in some *Methanobrevibacter* spp..

**_M. igneus_ and _M. jannaschii_ are high-pressure CH₄ cell factories.** High-pressure cultivation of methanogens offers an opportunity to improve the gas transfer rate of substrate gases into the liquid phase. In order to investigate the gas conversion kinetics and the barotolerance of applied methanogens, high-pressure experiments were designed to examine the $MER_{max}$ and conversion kinetics, including the turnover rate and the maximum conversion rate ($k_{min}/bar\ h^{-1}$) at a hyperbaric pressure of 10 and 50 bar in the simultaneous bioreactor system (SBRS)[39].

Four subsequent repetitive closed batch (RCB) experiments were performed by flushing and replenishing the SBRS headspace after reaching turnover$_{max}$ (Supplementary Figs. S9 and S10). After adaptation to hyperbaric conditions in RCB1, ten of the 14 prioritized methanogens achieved $MER_{max}$ and turnover rates in RCB2 (Fig. 3 and Supplementary Fig. S10). The MERs of *M. jannaschii* indicated a putative liquid limitation already in RCB2. Seven methanogens showed similar MERs and turnover rates in RCB3 and RCB4, probably as a result of limitation of the liquid substrates. Half of the tested strains showed an enhanced MER, when comparing 2 bar HFG and 10 bar RCB experiments (Figs. 2

and 3). In general, a fivefold higher pressure lead to an average MER increase of $2.2 \pm 0.9$ mmol $L^{-1} h^{-1}$. Some methanogens showed an average MER fold decrease of $0.6 \pm 0.1$ mmol $L^{-1} h^{-1}$, which might indicate a pressure sensitivity or could have been due to a potentially low pH present at hyperbaric conditions, since higher pressure in the cultivation vessel results in a higher soluble $CO_2$, which lowers the pH[40].

The highest MERs of $6.14 \pm 0.12$ mmol $L^{-1} h^{-1}$ and $4.39 \pm 0.41$ mmol $L^{-1} h^{-1}$ were achieved by *M. jannaschii* in RCB2 and RCB3, respectively, even without growth medium optimization. Besides MER and turnover rate, the parameter $k_{min}$ (Supplementary Fig. S11), derived from the maximum negative slope of the pressure curves (Supplementary Fig. S9), indicates the time point of $MER_{max}$. $k_{min}$ was therefore used to unambiguously identify the most productive strains and prioritize them for the subsequent 50 bar cultivations. These strains were *M. jannaschii*, *M. igneus*, *M. villosus* (using 282c 18_E medium), and *M. marburgensis*. The $CO_2$-BMP model organism *M. marburgensis* was successfully cultivated at 50 bar without facing liquid limitations (Fig. 4). *M. thermaggregans*, which is a high CH₄ productivity strain in fed-batch cultivation mode[13], did not grow at 50 bar (Supplementary Fig. S12) and *M. villosus* and *M. igneus* showed a decrease of MER and turnover rate directly in RCB2 (Fig. 4 and Supplementary Fig. S13), indicating a liquid limitation or sensitivity toward low pH, putatively caused by hyperbaric cultivation conditions. *M. jannaschii* did not fully convert $H_2/CO_2$ in RCB2 (Supplementary Fig. S12). Therefore, the medium for *Methanocaldococcus* spp. requires improvement, which should be based on a spectrophotometric analysis of quantities and quality of trace element

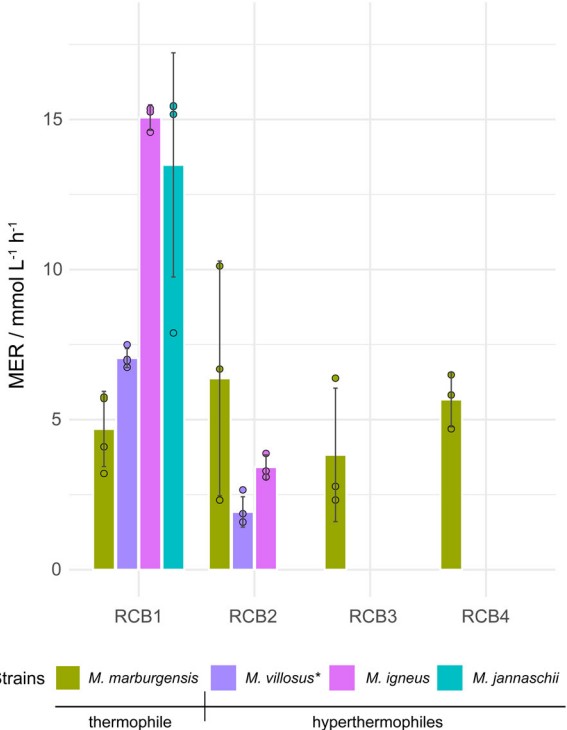

**Fig. 4 Results of RCB cultivations of thermophilic and hyperthermophilic methanogens in the SBRS at 50 bar.** The RCB cultivations were performed in quadruplicates with four runs RCB1, RCB2, RCB3, and RCB4. Mean and standard deviation of MER / mmol L$^{-1}$h$^{-1}$ is shown. Thermophile: *Methanothermobacter marburgensis* Marburg (65 °C, MM medium); hyperthermophiles: *Methanocaldococcus villosus* KIN24-T80* (80 °C, 282c18_E medium), *Methanotorris igneus* Kol 5 (85 °C, 282c 30 medium), and *Methanocaldococcus jannaschii* JAL-1 (80 °C, 282c 30 medium).

species and basal medium ingredients consumed during medium development[41,42]. Although the *Methanocaldococcus* medium needs further improvement, *M. villosus* and *M. igneus* did not show a lag phase at 50 bar. However, *M. villosus* directly started to convert H$_2$/CO$_2$ exponentially from the beginning of the experiment, compared to *M. igneus*, which showed linear, thus liquid-limited or low pH-retarded growth, directly from the onset of the cultivation (Supplementary Fig. S12). *M. igneus* comprised a threefold higher MER and *M. villosus* comprised a twofold higher $k_{min}$ compared to *M. marburgensis* at 50 bar, when comparing the performance during RCB1 (Fig. 4 and Supplementary Fig. S11). *M. igneus* exhibited the highest MER of 15.1 ± 0.4 mmol L$^{-1}$h$^{-1}$ and turnover rate of 4.8 ± 0.2 h$^{-1}$ in the 50 bar cultivation experiments (Fig. 4 and Supplementary Fig. S13), however we quantified a lower $k_{min}$ for *M. villosus* than for *M. igneus*. Furthermore, *M. igneus* and *M. jannaschii* exhibited the highest MER$_{max}$ (Supplementary Fig. S14). These results show that a nutrient limitation occurred and/or insufficient catalytically active biomass was present, meaning that CH$_4$ production was operated at qCH$_{4,max}$[12,20,30]. The fold increase of gaseous substrate in the media was found not to be proportional with the CH$_4$ productivity of the methanogens. The 25-fold higher gaseous substrate presence in the media, compared to a 2 bar cultivation, lead to an average MER increase of 2.92 ± 0.43 mmol L$^{-1}$h$^{-1}$. The time until full conversion of H$_2$/CO$_2$ at 50 bar was on average increased compared to 10 bar RCBs, by 2.5-fold (*M. marburgensis*, 67.79 ± 6.24 h), 2-fold (*M. villosus*, 43.21 h), and 5 h (*M. jannaschii*, 23.61 h). In contrast, *M. igneus* was 2 h (18.72 h) faster to reach full conversion under 50 bar RCBs compared to 10 bar RCBs.

## Discussion

Pure culture CO$_2$-BMP is regarded as a key technology combining chemical energy storage, CO$_2$ utilization and biofuel production. Within CO$_2$-BMP, methanogens are employed as autobiocatalytic CH$_4$ cell factories. Thus, we aimed to identify and characterize the highest performing CH$_4$ cell factories. This up to now unprecedented quantitative comparative physiological, bioinformatic, and biotechnological analysis provides a comprehensive view on growth and CH$_4$ production kinetics, essential nutritional components and barotolerance of 80 methanogens. The quantitative analysis of axenic methanogenic cultures enabled the identification of high performing cell factories for CH$_4$ production (high qCH$_4$) with a high maximum specific growth rate ($\mu_{max}$), straightforward cultivation methods (in terms of sterility, media demand, reproducibility), and tolerance to hyperbaric cultivation conditions.

Psychrophilic methanogens reached a rather low OD$_{max}$ < 0.2 in this study. This could be explained by the fact that psychrophilic microbes have in general a slower metabolism or a longer doubling time compared to microorganisms that grow at higher temperatures[43–45]. The heterogeneous growth pattern of mesophilic methanogens on complex and defined media could be explained by their ecological and phylogenetic heterogeneity. Although high biomass concentrations are often linked to growth on complex medium, highest productive methanogens do not necessarily require complex medium to reach a high OD. The highest CH$_4$ productivities were achieved by Methanococci, and especially by *Methanocaldococcus* spp. and *Methanotorris* sp. which exhibited higher conversions and CH$_4$ production kinetics (Fig. 2 and Supplementary Fig. S7) than thermophilic methanogens belonging to Methanobacteria.

Methanococci were shown to possess a faster metabolism, indicated by higher CH$_4$ production kinetics, possibly due to the usage of [NiFeSe]-hydrogenases for H$_2$ oxidation. Instead of using [NiFe]-hydrogenases for the oxidation of H$_2$ as *Methanothermobacter* spp. (F$_{420}$-reducing hydrogenase Frh and F$_{420}$-nonreducing hydrogenase Mvh), *Methanococcus* spp. and *Methanocaldococcus* spp. use [NiFeSe]-hydrogenases (F$_{420}$-reducing hydrogenase Fru and F$_{420}$-nonreducing hydrogenase Vhu) that display much higher catalytic activities[46,47]. Additionally, *Methanocaldococcus* spp. do not harbor selenium-free hydrogenases[46,48,49]. The catalytic activity of [NiFeSe]-hydrogenases is greatly increased compared to [NiFe]-hydrogenases. Vhu of *M. voltae* showed a catalytic activity of 43,540 U mg$^{-1}$[47], whereas Mvh of *M. marburgensis* indicated a catalytic activity of 1600 U mg$^{-1}$[50].

Our results reveal that methanogens, which showed the highest turnover rates and MERs, were covered with an S-layer. S-layer proteins can be positively or negatively charged, and it has been shown that charged S-layers enhance diffusion through the membrane[51]. The cell envelope of *M. kandleri* is known to be covered with an S-layer[36], although no S-layer motif was found during our UniProtKB search. Therefore, one could hypothesize that the S-layer proteins present on *M. kandleri* are characteristic for this phylogenetic group. Furthermore, our bioinformatic analysis of MCRα revealed that all highly productive prioritized methanogens harbor the Tyr$^{α444}$ Gly$^{α445}$ Tyr$^{α446}$ amino acid motif and belong to Class I methanogens (Supplementary Fig. S8).

Among the amino acids, especially cysteine is a required media supplement for certain methanogens (Supplementary Table S2). Compared to the prioritized Methanobacteria, hyperthermophilic Methanococci have a necessity of cysteine in the cultivation media, although Class I methanogens (Methanobacteriales, Methanococcales, and Methanopyrales) use primarily sulfide and not cysteine as sulfur source, such as Class II methanogens[52]. The

cysteine requirement of hyperthermophilic Methanococci in the medium could be linked to the usage of cysteine via cysteine desulphidase (CDD) for $H_2S$, $NH_4^+$, $H^+$, and pyruvate production[53], the production of cysteine via the t-RNA dependent pathway (SepRS/SepCysS)[54,55], and absence of cysteine desulphurase (CSD)[52,53,55] (Supplementary Table S2). Besides that, CDD seems to be associated with the sulfur transfer for Fe-S cluster biosynthesis[55–57]. In case of M. fervidus, where CSD was found to be expressed and CDD had not been (Supplementary Table S2), cysteine might have a key function in tolerating elevated temperatures[58].

Besides the nutritional demand of methanogens regarding cysteine, the TES that is used in a medium plays an important role in the biocatalytic activity. The trace element composition of a medium should mimic the heavy metal composition and respective concentrations present at the isolation spot, but might need to be optimized for meeting a biotechnological purpose. Based on our findings during the multivariate comparative analyses, methanogens that were cultivated on a medium with a rich-TES composition (TES1, TES2, TES4, and TES5) require additional cysteine or vitamins in the growth medium. Growth on a defined medium including a minimal/optimized TES (TES3), without cysteine or vitamins, was just possible for certain groups of methanogens, such as some Methanobacteria and M. maripaludis (cluster 3 in Supplementary Data 1, Table S1, and Supplementary Fig. S6). Vice versa, strains that grow best on media with a rich-TES composition, cysteine or vitamin addition indicated poor growth and $CH_4$ productivity on a medium with a minimal TES (TES3, MM medium), even with cysteine and/or vitamins also added. This leads to the conclusion that the combination of a rich TES and the addition of cysteine and/or vitamins is essential for the tested hyperthermophilic methanogens to exhibit high MERs.

We obtained the highest conversion and $CH_4$ production kinetics under hyperthermophilic and hyperbaric conditions. $H_2$ solubility at hyperbaric pressure of 10 or 50 bar leads to a 5- or 25-times higher substrate availability in the medium, compared to a cultivation at 2 bar. Therefore, adaptations to hyperbaric conditions, liquid limitation, and the suitability of the cultivation medium for high-pressure bioreactor cultivations can be studied if the experimental set-up is designed accordingly[12,20,24,30]. Instead of achieving a 5- and 25-fold productivity increase at 10 and 50 bar RCB experiments, an average of two- and threefold productivity increase was achieved, respectively. This might be due to cell envelope characteristics of the investigated methanogens and/or corresponding low pH[40], lipid composition, limitation of conversion kinetics by a liquid nutrient, or not enough available catalytically active biomass (biomass limitation) to instantly convert the additionally available gas, which could also be a result from a liquid limitation or natural borders if the culture is growing at $\mu_{max}$.

The $CH_4$ productivity pattern between RCB1 and RCB2 at 10 and 50 bar (Fig. 3 and M. marburgensis in Fig. 4) could be an adaptation response to hyperbaric cultivation conditions. However, the tested thermophilic and hyperthermophilic Methanococci have a different core lipid composition (archaeol, macrocyclic archaeol, and tetraether lipids) than Methanobacteria (archaeol and tetraether lipids). Strains from both orders increase the percentage of tetraethers under challenging growth conditions (Supplementary Table S3). M. jannaschii decreases archaeol and increases the percentage of tetraether lipids with increasing temperature[59], or temperature and pressure[60], while M. marburgensis increases tetraether lipids (GDGT-0), when growing with detergents[61]. Moreover, M. okinawensis increases tetraether lipids (GMGT-0, GMGT-0', and GDGT-0) and decreases archaeol upon addition of high amounts of inhibitors, such as

ammonium chloride and/or methanol, except for formaldehyde, which leads to an increase of archaeol[7,8].

At 10 bar, putative liquid limitation or biomass limitation occurred during RCB3 and RCB4 (M. marburgensis, M. thermautotrophicus, M. jannaschii, M. vulcanius, M. villosus, M. igneus, and M. kandleri) (Fig. 3 and Supplementary Fig. S9). However, at 50 bar putative liquid limitations arose right after RCB1 for Methanocaldococcus spp. and during RCB3 for M. marburgensis. Our findings indicate that just M. marburgensis is growing on a well-optimized medium (MM medium)[30]. The growth media (282c 18 or 282c 30) for Methanocaldococcus spp. would need to be adapted for hyperbaric applications. Although 282-based media were not yet designed for cultivations at 50 bar, the time for full conversion of $H_2/CO_2$ was not affected in the cases of M. igneus and M. jannaschii, which did not show any retardation in $CH_4$ production during 50 bar cultivations. Perhaps these strains could be tested at higher pressure conditions, such as M. okinawensis, which showed $CH_4$ production up to 90 bar[40]. Methanocaldococcus spp. exhibited higher specific growth rates than M. marburgensis (Supplementary Data 1 and Table S1), and thus liquid limitation occurs faster. Besides that, the metabolism of M. marburgensis is slower compared to Methanocaldococcus spp., indicated by the lower $k_{min}$ values of Methanocaldococcus spp. (Supplementary Fig. S11). Therefore, the liquid limitation in our setup might not have had a strong effect.

This study on high-pressure biological $CH_4$ production in pure culture is a cornerstone of the emerging research and development field of Archaea Biotechnology[19]. The systematic assessment indicated that the high-performance strains belong to Class I methanogens. Hyperthermophilic Methanococci are high-pressure $CH_4$ production cell factories and the addition of cysteine and a rich TES in the media are essential for efficient growth of these Methanococci. Therefore, we propose to perform bioprocess development utilizing M. igneus and M. jannaschii to develop these organisms into high-pressure $CH_4$ cell factories. Moreover, methanogens that exhibited the highest turnover rates and MERs are covered with S-layers, and the amino acid motif $Tyr^{\alpha444}$ $Gly^{\alpha445}$ $Tyr^{\alpha446}$ in the alpha subunit of MCR is present in all high-performance methanogens. This analysis sets the foundation for a future high-pressure bioprocess optimization endeavor with the identified hyperthermophilic $CH_4$ cell factories. The autobiocatalytic activity of hyperthermophlic, autotrophic, hydrogenotrophic methanogens could therefore be employed for balancing the power grid system (energy storage) or to biologically depressurize $H_2$ and/or $CO_2$ containing emission flue gasses to $CH_4$ via the $CO_2$-BMP process. High-pressure biological $CH_4$ production in pure culture could provide a purposeful route for the transition to an independent carbon-free or low-carbon energy bioeconomy.

## Methods

**Strains**. All screening experiments including HFG (closed batch up to 2 bar) were performed with the methanogenic archaeal strains listed in Fig. 1. Methanogens were obtained from the Deutsche Sammlung für Mikroorganismen und Zellkulturen GmbH (DSMZ) (Braunschweig, Germany). High-pressure experiments were performed with selected strains in the SRBR in closed batch mode at 10 and 50 bar (Figs. 3 and 4).

**Chemicals**. $CO_2$ (99.995 Vol.-%), $H_2$ (99.999 Vol.-%), and $H_2/CO_2$ (80 Vol.-% $H_2$ in $CO_2$) were obtained from Air Liquide (Air Liquide GmbH, Schwechat, Austria). The $H_2/CO_2$ mixture (80 Vol.-% $H_2$ and Vol.-% $CO_2$) for high-pressure cultivations was obtained from Linde Gas (Linde Gas GmbH, Wels, Austria). All other chemicals were of highest available grade.

**Media**. Considering the nutritional requirements of the screened strains, several media were used to cultivate methanogenic archaeal strains, as they are SAB medium[28], McN medium[31], Medium 6[29], DSMZ medium (141, 141b, 141c, 282,

203, 511), *Methanothermobacter marburgensis* medium (MM)[30] and MM medium with 15 or 30 g of NaCl (MM15, MM30). Some media were modified to test specific nutritional requirements (203c, 203-c, 511-v, 282c 0, 282c 18, 282c 18_E, 282c 30, 282-c 30, MM15c, MM15v, MM30c). The addition of "c" to the medium designation indicates the presence of cysteine in the medium, whereas "-c or -v" indicates the omission of cysteine or vitamins, respectively; the succeeding numbers refer to the amount of NaCl provided to the medium. Every media or solution was prepared with ultrapure $H_2O$ Milli-Q® if not stated differently. To ensure growth in a respective medium and exclude any stimulative effects of medium contained in the inoculum, methanogens were grown in two passages or two times washed with fresh medium. Before inoculation on minimal medium, inocula were washed by centrifugation following by the removal of the supernatant, adding 1 mL of minimal medium and resuspending the pellet, centrifugation (10 min, 13,000 rpm), discarding the supernatant and resuspending the pellet in 1 mL of minimal medium. The exact media compositions are listed in Supplementary Information.

**Multivariate analysis of cultivation conditions**. Growth and $CH_4$ productivity of 80 methanogenic archaea were screened in closed batch cultivation mode up to 2 bar relative to atmospheric pressure, 2 barg, in an anaerobic atmosphere consisting out of 80% $H_2$ in $CO_2$ (4:1). For simplicity, all pressure assignments are described as 2, 10, or 50 bar relative pressure. The optimal growth temperatures of the tested methanogens range from 15 to 98 °C. The tested methanogens are classified as psychrophiles including psychrotolerant methanogens (15–30 °C), mesophiles (30–37 °C), thermophiles (40–70 °C), and hyperthermophiles (80–98 °C). Methanogenic strains were grown in 120 mL serum bottles (crimp neck vial, VWR International, Pennsylvania, USA) in chemically defined media (see "Media" section). After autoclaving, media filled bottles (autoclave Systec VX-120, Systec GmbH, Linden, Germany), inoculation was performed inside an anaerobic chamber (Coy Laboratory Products, Grass Lake, USA). Thereafter, bottles were pressurized with a $H_2/CO_2$ gas mixture (80% $H_2$ in $CO_2$) at 2 bar as previously described[62]. A sterile gaseous substrate supply requires the usage of sterile syringe filters (w/0.2c μm cellulose, 514-0061, VWR International, USA) and disposable hypodermic needles (Gr 14, 0.60 × 30 mm, 23 G × 1 1/4″, RX129.1, Braun, Germany). The gas phase was flushed by an insertion of a second needle at regular intervals for 2–4 s. After pressurizing, bottles were incubated in water baths (orbital shaking, water bath 1083, GFL Gesellschaft für Labortechnik mbH, Germany) or air incubators (100 rpm, Labwit incubators, Labwit Scientific Pty. Ltd, Australia) according to the optimal growth temperature of the respective cultivated strains. Bottles were taken out of the incubator and cooled down or heated up to room temperature. Thereafter, pressure and $OD_{578 \, nm}$ measurements (liquid samples of 0.7 mL were taken) were performed to monitor the cultivation. After the measurements, bottles were flushed, repressurized, and incubated again at microbial-specific cultivation temperatures.

**High frequency gassing experiments**. The experimental set-up of HFG followed the procedure described above, except for the gassing frequency, which was increased to twice a day. In total 14 strains, three mesophilic methanogens grown at 37 °C (*M. maripaludis* S2, *M. palustre* F, *Methanobacterium subterraneum* A8p), five thermophilic methanogenic archaea cultivated at 65 °C (*M. marburgensis* Marburg, *M. thermophilus* M, *M. thermaggregans*, *M. thermautotrophicus* DeltaH, *M. okinawensis* IH1), and six hyperthermophilic methanogens (*M. jannaschii* JAL-1 (80 °C), *M. vulcanius* M7 (80 °C), *M. villosus* KIN24-T80 (80 °C), *M. igneus* Kol 5 (80 °C), *M. fervidus* H9 (80 °C), *M. kandleri* AV19 (98 °C)) were selected for HFG.

**Dormancy study**. Before methanogens reached the stationary growth phase, they were put into dormancy state in a 4 °C room or in a −80 °C freezer (Thermo Scientific™ TSU™ Series −86 °C Upright Ultra-Low Temperature Freezers, Thermo Fisher Scientific, USA). For the −80 °C dormancy study, cryostocks were used (800 μL culture and 200 μL 50% (v/v) glycerol). After dormancy at 4 or −80 °C, strains were inoculated into fresh medium. In case of strains that were kept at 4 °C, an aliquot of 1 mL was used as inoculum. The 1 mL −80 °C cryostocks were thawed and used as inoculum after removing the glycerol by centrifugation (10 min, 13,000 rpm). The respective dormancy periods are listed in Supplementary Table S1.

**Analysis of growth and productivity**. During all 2 bar screening experiments including HFG, growth and $CH_4$ formation was examined by OD and pressure measurements. Growth was monitored via offline OD measurements at 578 nm ($OD_{578 \, nm}$) by using a spectrophotometer (DU800, Beckman Coulter, California, USA). Before every $OD_{578 \, nm}$ measurement, the sample was vortexed (Vortex Mixer MX-S, Biologix Group Limited, China). In total, 0.7 mL of the culture was sampled at regular intervals for $OD_{578 \, nm}$ determinations. $CH_4$ production capacity was investigated through headspace pressure measurements of serum bottles in regular intervals using a digital manometer (LEO1-Ei, -1/3 bar relative, Keller, Germany)[27]. Produced $CH_4$ was replaced by discontinuous gassing with $H_2/CO_2$ in regular intervals.

**Data analysis**. Two heatmaps are shown in Fig. 1, illustrating max. growth via $OD_{max}$, by measuring at 578 nm, and max. volumetric $CH_4$ productivity,

depicted as $MER_{max}$ / mmol $L^{-1}$ $h^{-1}$. The heatmaps showing turnover$_{max}$ / % (Supplementary Fig. S2) and biomass increase rate (Supplementary Fig. S1) can be found in Supplementary Information. The biomass increase rate depicts the average value of all biomass increase rates of a specific strain during the cultivation and is calculated according to Eq. (1). Figure 2 shows two boxplots, max. volumetric $CH_4$ productivity as MER / mmol $L^{-1}$ $h^{-1}$ and the biomass increase rate. The corresponding boxplot illustrating the turnover rate / $h^{-1}$ is shown in the Supplementary file (Supplementary Fig. S7). All data points were included into the boxplots. The heatmaps and the boxplots were generated using Rstudio Version 1.1.463 – © 2009-2018 RStudio, Inc. The R package ggplot2[63] was used. The graphical design was refined using Illustrator CS6 (Adobe Systems Inc., USA).

$$\frac{OD_{max} \cdot \mu}{OD \cdot \mu_{average}} / - \tag{1}$$

**Correlation between nutritional demand and productivity**. To correlate the nutritional demand of methanogens with their associate growth, substrate conversion, and productivity on respective media, a standardized principal components analysis (PCA) followed by a k-means clustering was performed. Clustering was performed on $OD_{max}$, turnover$_{max}$, $MER_{max}$, and the combination of those with medium-associated components such as salt, sulfate, sulfur, ammonium, phosphate, and cysteine concentrations. This analysis was then linked to medium-based information (trace elements solution, VS, the addition of yeast, peptone, or cysteine) and strain-specific characteristics like taxonomy and cultivation temperature. After collecting the data, missing values got imputed via PCA imputation using the R package missMDA[64], followed by a normalization of the data using the stats package[65]. Thereafter the PCA (stats[65]) was performed, followed by the k-means clustering (stats[65]) using the first two components. The within cluster sum of squares accounted 85.5% for $OD_{max}$, 93.2% for $MER_{max}$, 91.8% for turnover$_{max}$, and 86.3% for the combination of these variables. The following R packages were applied during the analysis: R packages ggplot2[66], missMDA[64], FactoMineR[63,66,67], and stats[65]. The biplots were generated using Rstudio Version 1.1.463 – © 2009-2018 RStudio, Inc. The graphical design was refined using Illustrator CS6 (Adobe Systems Inc., USA).

**Cell envelope and S-layer composition of prioritized methanogens**. Cell envelop structures including core lipid composition and putative S-layer presence on prioritized methanogens were investigated via literature research. Furthermore, a bioinformatic screen on the UniProt Knowledgebase (UniProtKB)[68] regarding the presence of S-layer on tested methanogenic strains was conducted. The combination of strain-specific designation and "S-layer protein, glycoprotein, or glyco protein" were used as query terms. Additional information about protein family classification (Interpro) and functional regions/domain of the protein are indicated via Pfam[69].

**Amino acid conservation of curtail interacting partners within methyl-coenzyme M reductase I, subunit alpha**. Protein sequences were obtained using the protein–protein BLAST (blastp)[70,71]. The Reference Sequence (RefSeq) collection was used as sequence database for the blastp (version January 2020). MCR I, subunit alpha from *Methanothermobacter marburgensis* Marburg (GenBank: ADL59127.1) was used as a query. The default algorithm parameters were chosen (scoring matrix BLOSUM62) besides the max. target sequences which was increased to 500. Protein sequences from methanogens that were investigated during this study and were not obtained through blastp were afterwards added to the blastp sequences. Missing protein sequences were downloaded from UniprotKB[68]. Thereafter, protein sequences were aligned with the multiple sequence alignment tool Clustal Omega[72] applying the default settings. The download of amino acid sequences and the following analysis was performed in February 2020. The alignment was illustrated using Jalview version 2.10.5[73]. The graphical design was refined with Illustrator CS6 (Adobe Systems Inc., USA).

**High-pressure SRBS cultivation of methanogens**. An experimental design was developed to examine $H_2/CO_2$ conversion kinetics through online pressure measurements and to identify the most productive methanogens at a hyperbaric relative pressure of 10 and 50 bar. Further, possible liquid limitations and kinetic stability of $CH_4$ production of prioritized methanogens at 10 bar was examined. The SBRS consisted of four identical cultivation vessels (160 mL), suitable for investigation of microbial activity at pressures up to 50 bar and temperatures up to 145 °C[39]. All pressure assignments are given in bar and described as relative pressure (10 and 50 bar relative to atmospheric pressure). Pressure within the bioreactors (R1, R2, R3, and R4) was accurately measured with online pressure sensors (Pressure Transducers and Transmitters, Type: PTDVB0601B1C2, Parker, Cleveland, USA). Before bioreactors were pressurized, the gas inlet line pressure was set to 10 or 50 bar of $H_2/CO_2$ with an analogous manometer (WIKA Messgerätevertrieb Ursula Wiegand GmbH & Co. KG, Vienna, Austria, 0–60 bar)[39]. Thus, pressure was checked online and offline during the pressurization step. During the experiments, pressure was monitored with the online pressure sensing tool. The pressure sensor and the manometer were calibrated before installation. The accuracy of the heating jacket was also tested beforehand. The fact that each

experiment was performed in quadruplicates, strengthens the validity of the results. Some high-pressure experiments could not be investigated with all four bioreactors due to technical or biological failure. Since this was a screening approach, high-performance methanogens were further investigated and their growth and $CH_4$ production kinetics were then analyzed in detail.

**SBRS inoculation procedure**. After cleaning and autoclaving the SBRS, an anaerobic environment in each bioreactor vessel was established, followed by setting the cultivation temperature, respectively, to the strain's optimal growth temperature based on DSMZ data. Before combining medium, supplements, and culture in a vial (100 mL crimp neck vial, Macherey-Nagel GmbH & Co. KG, Germany), the culture was reactivated for 10 min via flushing with 1 bar $H_2/CO_2$. Based on the used medium, supplement solutions like $NaHCO_3$, 0.5 mol $L^{-1}$ $Na_2S$, L-Cysteine-$HCl \cdot H_2O$, and vitamins (Wolf's VS, see medium 141) were added to the medium right before inoculation. The inoculum (medium, supplements, and culture) was transferred into the bioreactor[39]. Thereafter an appropriate incubation pressure (10 or 50 bar, depending on the experiment) was adjusted. The RCB set-up included three repressurization steps either at 10 or 50 bar, respectively. After a total gas conversion, a gas sample from each bioreactor was taken for gas composition analysis via gas chromatography. Full conversion is achieved if 2 bar (10 bar) or 10 bar (50 bar), i.e. one-fifth of the initial pressure, of residual gas in the bioreactors R1, R2, R3, and R4 is present, respectively, to production of $CH_4$ ($4 H_2 + CO_2 \rightarrow CH_4 + 2 H_2O$) to avoid a driving force limitation. Before repressurization, the residual pressure was released. The following strains were investigated for 10 bar RCB: mesophiles 37 °C: *M. maripaludis* S2 (MCN medium), *M. palustre* F (282c 0 medium), *M. subterraneum* A8p (MM medium); thermophiles 65 °C: *M. marburgensis* Marburg (MM medium), *M. thermophilus* M (was not tested), *M. thermaggregans* (MM medium), *M. thermautotrophicus* DeltaH (MM medium), *M. okinawensis* IH1 (282c 30 medium); hyperthermophiles 80 °C: *M. jannaschii* JAL-1 (282c 30 medium), *M. vulcanius* M7 (282c 30 medium), *M. villosus* KIN24-T80 (282c 18 and 282c 18_E medium), *M. fervidus* H9 (MMc15 medium); hyperthermophiles 85 and 98 °C: *M. igneus* Kol 5 (282c 30 medium), *M. kandleri* AV19 (511 medium). *M. maripaludis*: in RCB1, R2 was leaking at the beginning and fixed after taking note of the leakage. After 80 h in RCB3 no growth was observed, thus RCB3 was stopped. *M. subterraneum*: R2 was not functional during all RCB runs. After recognizing the leakage on the following day, the pressure decreased to 3.3 bar. Subsequently R2 bioreactor was repressurized and RCB2 was started. R2 in RCB3 was again leaking. The problem was fixed before starting RCB4. *M. thermautotrophicus*: the pressure curve of R1 in RCB3 (100 h) did not follow the trend of the others, thus it is was not shown and not repressurized in RCB4. R1 was not working for cultivations of *M. jannaschii*, *M. vulcanius*, and *M. igneus*. *M. fervidus*: the performance of R1 was not comparable to the others in RCB1 and therefore not repressurized for following RCB runs. *M. kandleri*: after RCB3, R1 was not working properly and therefore not repressurized.

The following strains were investigated in 50 bar RCBs: *M. marburgensis* Marburg, *M. thermaggregans*, *M. villosus* KIN24-T80\*, *M. igneus* Kol 5, and *M. jannaschii* JAL-1. After RCB2 cultivations of *M. villosus* KIN24-T80\*, *M. igneus* Kol 5, and *M. jannaschii* JAL-1 the experiments were stopped due to much lower $CH_4$ productivity compared to RCB1. *M. thermaggregans*: after 120 h of cultivation no growth was detected, thus RCB1 was stopped.

**Analysis of biomass, $CH_4$ productivity, and head space gas composition (SBRS)**. During all RCB runs at 10 and 50 bar, growth and $CH_4$ formation were examined via offline cell dry weight analysis, online pressure measurements, and offline headspace gas determinations. Growth was determined via offline cell dry weight analysis by using a centrifuge (tabletop centrifuge Heraeus Megafuge 1.0 R, Thermo Electron Corporation, Massachusetts, USA) to pellet the harvested biomass. Biomass was centrifuged for 15 min at 3500 rpm, followed by drying the wet biomass at 105 °C overnight (Heraeus drying cabinet model T 5050, Heraeus, Hanau, Germany). Pressure drop, which corresponds to $CH_4$ production, was monitored by online pressure sensors. After total gas conversion, gas samples were taken via headspace vials (10 mL headspace vial, Schmidling Labor + Service GmbH, Switzerland) which were crimped with crimp caps (crimp cap with bore hole, Carl Roth, Germany) and vacuumed for 5 min before usage. The $CH_4$ off-gas concentration ($CH_4$ / Vol.-%) in the gas samples are analyzed with a gas chromatograph (Trace GC Ultra 2000, Thermo Fisher Scientific Inc., US) equipped with a thermal conductivity detector. Chromatographic separation was executed on a Carbonex-1000 packed column (10 m, 3/8″). Helium used as carrier gas with a constant pressure of 2.35 bar and a split flow of 90/10. A representative gas sample with a volume of 1 mL was injected. Following GC parameters were chosen for the analysis: inlet heater 150 °C, detector 200 °C, oven initial temperature 35 °C hold for 5 min, temperature raising rate of 20 °C $min^{-1}$ to 225 °C (hold for 10 min) at final temperature.

**Data analysis of SBRS 10 and 50 bar cultivations**. To elucidate the $CH_4$ production kinetics from cultivated methanogenic strains, the following variables were calculated: methane evolution rate calculated with GC data $MER_{GC}$ / mmol $L^{-1} h^{-1}$ or pressure data $MER_{pressure}$ / mmol $L^{-1} h^{-1}$, carbon uptake rate CUR / mmol $L^{-1} h^{-1}$, hydrogen uptake rate HUR / mmol $L^{-1} h^{-1}$, maximum conversion rate $k_{min}$ / bar $h^{-1}$.

MER was calculated either by using $CH_4$ concentration obtained from GC measurements or through the integration of the recorded curve for the online pressure probe in every bioreactor. Data collection was performed by a data acquisition unit (USB-2019, ICP DAS-EUROPE GmbH, Germany) and recorded via LabVIEW (National Instruments, Austin, USA). During cultivation the record interval was set to 5 min. All calculations were performed by using the program Origin 2019 (Originlab Corporation, USA). After selecting a proper record interval (30 min) for data analysis, the $CH_4$ production kinetics (MER, turnover rate, $k_{min}$) were calculated, while neglecting biomass formation[39,62]. These MER values were used to determine the respective point in time where MER reached its maximum ($MER_{max}$). $k_{min}$ indicates the highest slope in the curve, which reflects the point of highest turnover, showing the time point of $MER_{max}$. $MER_{total}$ indicates the MER value over the total experimental time including all data points. MER values got smoothed and plotted over time. To determine the lag and the stationary phase during cultivation and neglect these data points, an integration over the obtained curve was performed. The integration start ($x_{start}$) and end ($x_{end}$) points of the curves were identified as follows: $x_{start}$ and $x_{end}$ are intersection points, which were elucidated by shifting the x-axis to low points of the curve (minimum $x = 0.1$). The difference of these points reflects the time period $\Delta t$ of microbial growth phases. $MER_{global}$ was determined by dividing the calculated area by $\Delta t$[33]. Finally, the program Origin indicated the associated pressure data for the $x_{start}$ and $x_{end}$ values. The associated pressure data were subsequently used for calculating of $MER_{GC}$, $MER_{pressure}$, CUR, HUR, and $k_{min}$. The barplots were generated using Rstudio Version 1.1.463 – © 2009-2018 RStudio, Inc.. The R package ggplot2 was used[66]. The graphical design was refined using Illustrator CS6 (Adobe Systems Inc., California, USA).

**Statistics and reproducibility**. All information on statistics and reproducibility of the experiments is provided in the respective "Materials and methods" sections or in the Supplementary Information, Supplementary Data 1 and Data 2. The 2 bar multivariate screening experiments were performed with 80 methanogens with an ranging optimal growth temperature between 15 and 98 °C. The tested methanogens are classified as psychrophiles including psychrotolerant methanogens (15–30 °C), mesophiles (30–37 °C), thermophiles (40–70 °C), and hyperthermophiles (80–98 °C). The multivariate screening was followed by the HFG experiments, which were performed with 14 methanogens. For each closed batch cultivation (multivariate screening or HFG), three biological replicates (in some cases, two biological replicates) plus one negative control were used. Thereafter, 10 and 50 bar hyperbaric cultivations were performed. The 10 bar RCB cultivations were investigated with 13 methanogens. Followed by 50 bar RCB cultivations with four methanogens. Hyperbaric cultivations were performed in quadruplicates. In some cases, just three bioreactors were used, due to a non-functionality of one of the bioreactors of the SBRS.

**Reporting summary**. Further information on research design is available in the Nature Research Reporting Summary linked to this article.

## Data availability

All relevant data are available from the corresponding author upon request.

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

## Acknowledgements

Maria Wolfsgruber is acknowledged for assisting with high-pressure experiments. Dr. Lydia M.F. Baumann is acknowledged for helpful discussions. Dr. Melina Kerou is acknowledged for proofreading the manuscript. Greatly acknowledged is the Österreichische Forschungsförderungsgesellschaft (FFG) for funding the projects BioHyMe (grant 853615), Bioraffinerie (grant 854156), and NitroFix (grant 859293). Open access funding by the University of Vienna.

## Author contributions

L.-M.M.: writing paper draft, writing paper review and editing, closed batch cultivations, high-pressure cultivations, bioinformatic analysis, data analysis, preparation of figures. S.Z.: high-pressure cultivations, data analysis. P.P.: high-pressure cultivations, data analysis. S.B.: design of experiments, writing paper review and editing. A.H.S.: design of experiments, writing paper review and editing. B.R.: closed batch cultivations. T.S.: closed batch cultivations. R.-S.T.: closed batch cultivations, writing paper review and editing. C.P.: supervision of high-pressure experiments, supervision of data analysis, writing paper review and editing, funding acquisition. S.K.-M.R.R. closed batch cultivations, supervision of closed batch experiments, supervision of data analysis, supervision of bioinformatic analyses, writing paper draft, writing paper review and editing, funding acquisitions.

## Competing interests

S.B. and A.H.S. declare to have competing financial interests due to their employment in the Krajete GmbH. All other authors declare no competing interests.
