## [Peer Review File · Communications Biology]

Reviewers' comments:

Reviewer #1 (Remarks to the Author):

The manuscript presents very interesting new results. Below specific comments which should be considered.

1. The EU policy is to produce hydrogen as the fuel of the future due to the lack of CO₂ emissions. In contrast to biomethane, hydrogen is emission-free. The use of hydrogen from RES for CO₂ sequestration will seem pointless. It seems more reasonable to use hydrogen for temporary transformation into methane for its storage and transmission. Subsequently, its conversion to hydrogen would be analogous to the regasification of liquid methane to gas. One statement on page 3; line 63 - does not reflect the essence of the problem.
2. Page 3; lines 72-74 - there is no literature reference to the information for the statement "This metabolic capability is essential when it comes to organic matter degradation in anoxic environments with low sulphate, nitrate, and Mn (IV) or Fe (III) concentrations." Some publications indicated an increase in methane production with the addition of iron filings.
3. Page 5; lines 122-123 - the variables lack the control of dissolved CO₂ and H₂ concentration for the mentioned control of homogeneous or heterogeneous growth. With the use of such a variety of pressurized media, a wide variety of chemical reactions could occur.
4. Page 6; line 137 - subscript included in part of the sentence.
5. Page 9; Fig.1 - illegible figure, I propose to break the figure to increase its legibility.
6. Materials and methods mention the use of an H₂: CO₂ 4: 1 ratio. I did not find any information as to why this ratio was assumed. Bioresource Technology publication 233 (2017) 256–263 shows that only a higher ratio of 6: 1 to 10: 1 gives the best results for CO₂ conversion and CH₄ production. With such parameters, such rapid drops in pH would not have occurred, so that other species of methanogens could show a better result.
7. In the supplement S1 to FIG1 with the HTS screening experiment to select medium, the information about temperature is missing.
8. Authors should include supplement the protocols for testing the storage capacity of methanogen strains for continuation of research in the event of infection or contamination of experiments - page 10 verse 236-237.
9. Authors should include/crlarify supplement containing bioinformatic analysis protocols and proteomics demonstrating correlation of the presence of S surface proteins in the outer membrane with the increased ability of methanogens possessing them to a greater ability to reduce CO₂. Same Supplements regarding the demonstrated relationship between the specific set of amino acids Tyr α 444 Gly α 445 Tyr α 446 in the construction of methyl-coenzyme M reductase (α Mcr) proteins that are key for methanogens and increased conversion efficiency for archaea possessing it. Sets for all methanogens.
10. There are some minor typing errors.

Reviewer #2 (Remarks to the Author):

(See also attachment)

General comment

The manuscript assesses high pressure methane-generating cell factories for H₂/CO₂ conversion

using single cell cultures. 80 different species were evaluated for their performance ranging from psychrophilic to hyper thermophilic range and provides valuable insights on the effect of medium composition and elevated pressures as well as phylogenetic characteristics of high performing cell-factories. The manuscript provides comprehensive data and many insights that are expected to be valuable for the continuation technical development in this field as well as contributing to the fundamental knowledge of these methanogenic archaea. Overall the manuscript provide relevant and novel information that is expected to gain interest from both the academic community as well as being industrially relevant.

However, the results and discussion are difficult to follow and the English needs to be improved. It is also difficult in places to distinguish between what is result or conclusions originating from the current work and what is referring to previous work. In general, the introduction needs to be more focused and relevant for the current work and the results section needs to be divided in more sections with subheadings and better structured to facilitate the readability. Moreover, figures are difficult to read when printed out. Consider using bigger font and distinguishable bars. Sufficient information is given in the material and method section to reproduce the work. However, some issues related to the methods should be included in the discussion of the result as they may be contributing factors. Please refer to the specific comments below.

Please also consider mentioning the media optimization and the effect of the specific nutrient additions in the abstract since it constitutes a large part of the result content and elaborate more on the understanding these results with more references to previous work in both the introduction and discussion sections.

Specific comments

1. Line 35: CH₄ instead of CH4.
2. Line 35: In the abstract, the S-layer and MCR α are the only specific features mentioned from the bioinformatic study. Why is not the role of selenium in the hydrogenase enzyme mentioned?
3. Lines 49-61: This text does not introduce the current work and is not contributing to the understanding of the current work. Removing this would reduce the length of an already extensive introduction section and allow for the importance and relevance of the current work to be introduced to the reader.
4. Lines 79-84: In an AD process an undefined multi-strain consortium is more flexible and robust. In this text you just mentioned to positive sides for pure cultures. To be more convincing in your pure-culture strategy, it would be good to mention the positive argument of a diverse culture for example reduced risk of contamination. This would be interesting to put in relation to why for example growth and MES are relevant factors to study.
5. Lines 84-91: Similarly, to above you only mentioned the negative issues with Sabatier. By variegate the information and also mentioned for example lower rate of reaction of the biological system the essence of the current study would be better introduced.
6. Lines 97-98: Please provide reference. Moreover, here is the only place in the introduction where you mentioned high pressure. However, since this is a very important part of this study the reader needs more references to previous work, i.e what is previously known about pressure tolerance and H₂ availability and how the current study further increases the knowledge on this aspect. Similarly, the effect of nutrient addition and media composition should be mentioned. What is the current knowledge of these additives and the effect of rich v.s. mineral medium?
7. Line 122: You are using a high H₂/CO₂ ratio. Since this is a fundamental study the high ratio is accepted. However, since this ratio might not be industrially possible due to the expense of hydrogen, this issue should be better discussed in possible the introduction but clearly discussion

and conclusion sections.

8. Line 125 According to the equation 1 Line 552 the units for the biomass increase rate is incorrect it should be unitless. Please check and update in all places.

9. Line 115-248 this section is very long and difficult to follow. Please consider dividing it in more sections with subtitles to improve readability.

10. Line 132. Please mention MER also for thermophiles as you include this result for all other groups.

Fig 1 contains very relevant and important information but is very difficult to read. Please consider dividing it in more plots and increase the font size.

11. Line 230-239 please group the results on effect of dormancy and make a subheading or consider removing this part from the manuscript. It seems to me that the authors compares too many parameters (mesophilic and hyperthermophilic) as well as storage conditions 14 weeks and >1year, -80 and 4°C and with and without glycerol (the latter also including a extra treatment step of centrifugation that could also have an effect) to really make the conclusion concerning dormancy on lines 238-239.

12. Lines 253-255 Consider moving these lines to the introduction. See comment above

13. Line 282: consider adding a new subheading here to improve the readability of the manuscript.

14. Line 298: "M. kandleri should possess an S-layer,..." This statement is not clear. Are you refereeing to another study or is it your own observation. This needs to be better explained. Especially as the S-layer is lifted as a central result for the study in both abstract and introduction.

15. Line 358-359 The result could also be explained by sensitivity and therefore improving medium composition would not necessarily improve productivity as indicated in the sentence.

16. Line 401 Consider changing "have been" to "was" to clarify that the finding derived from this study on not refereeing to previous work. This is overall an issue with the text in general that should be rewritten keeping this aspect (weather the finding came from this study or it is referring to previous work) in mind.

17. Lines 401-407 Very long and difficult sentence that should be improved.

18. Line 418. Same comment as above consider replacing "should" with "are known to be" or similar to clarify that you refer to previous work.

19. Lines 510-513 The cooling and reheating of the cultures from thermophilic and hyperthermophilic conditions should affect the growth profile. The consequence of the cultivation should be included in the discussion of the results.

20. Line 552 Why did you use this equation has it previously been described. Please add reference.

21. Lines 604- 685 please provide the media composition in the supplement material and only add the essential information of the main differences between the composition in the main text.

22. Line 703 the accuracy of the HFG should be described. Did you do any calibration or other check of the accuracy of the machine this information would add to the quality of the manuscript. Also, pH was discussed in many places as an important factor correlated to pressure. Did you measure pH before or during cultivations?

Reviewer #3 (Remarks to the Author):

(See also attachment)

The experiment of this manuscript is well designed and presented. The results are also adequately

discussed. However, the reviewer suggests improving the English language and correcting several minor mistakes before publication.

Review report

General comment

The manuscript assesses high pressure methane-generating cell factories for H₂/CO₂ conversion using single cell cultures. 80 different species were evaluated for their performance ranging from psychrophilic to hyper thermophilic range and provides valuable insights on the effect of medium composition and elevated pressures as well as phylogenetic characteristics of high performing cell-factories. The manuscript provides comprehensive data and many insights that are expected to be valuable for the continuation technical development in this field as well as contributing to the fundamental knowledge of these methanogenic archaea. Overall the manuscript provides relevant and novel information that is expected to gain interest from both the academic community as well as being industrially relevant.

However, the results and discussion are difficult to follow and the English needs to be improved. It is also difficult in places to distinguish between what is result or conclusions originating from the current work and what is referring to previous work. In general, the introduction needs to be more focused and relevant for the current work and the results section needs to be divided in more sections with subheadings and better structured to facilitate the readability. Moreover, figures are difficult to read when printed out. Consider using bigger font and distinguishable bars. Sufficient information is given in the material and method section to reproduce the work. However, some issues related to the methods should be included in the discussion of the result as they may be contributing factors. Please refer to the specific comments below.

Please also consider mentioning the media optimization and the effect of the specific nutrient additions in the abstract since it constitutes a large part of the result content and elaborate more on the understanding these results with more references to previous work in both the introduction and discussion sections.

Specific comments

1. Line 35: CH₄ instead of CH4.
2. Line 35: In the abstract, the S-layer and MCR α are the only specific features mentioned from the bioinformatic study. Why is not the role of selenium in the hydrogenase enzyme mentioned?
3. Lines 49-61: This text does not introduce the current work and is not contributing to the understanding of the current work. Removing this would reduce the length of an already extensive introduction section and allow for the importance and relevance of the current work to be introduced to the reader.
4. Lines 79-84: In an AD process an undefined multi-strain consortium is more flexible and robust. In this text you just mentioned to positive sides for pure cultures. To be more convincing in your pure-culture strategy, it would be good to mention the positive argument of a diverse culture for example reduced risk of contamination. This would be interesting to put in relation to why for example growth and MES are relevant factors to study.

5. Lines 84-91: Similarly, to above you only mentioned the negative issues with Sabatier. By variegate the information and also mentioned for example lower rate of reaction of the biological system the essence of the current study would be better introduced.
6. Lines 97-98: Please provide reference. Moreover, here is the only place in the introduction where you mentioned high pressure. However, since this is a very important part of this study the reader needs more references to previous work, i.e what is previously known about pressure tolerance and H₂ availability and how the current study further increases the knowledge on this aspect. Similarly, the effect of nutrient addition and media composition should be mentioned. What is the current knowledge of these additives and the effect of rich v.s. mineral medium?
7. Line 122: You are using a high H₂/CO₂ ratio. Since this is a fundamental study the high ratio is accepted. However, since this ratio might not be industrially possible due to the expense of hydrogen, this issue should be better discussed in possible the introduction but clearly discussion and conclusion sections.
8. Line 125 According to the equation 1 Line 552 the units for the biomass increase rate is incorrect it should be unitless. Please check and update in all places.
9. Line 115-248 this section is very long and difficult to follow. Please consider dividing it in more sections with subtitles to improve readability.
10. Line 132. Please mention MER also for thermophiles as you include this result for all other groups.

Fig 1 contains very relevant and important information but is very difficult to read. Please consider diving it in more plots and increase the font size.

11. Line 230-239 please group the results on effect of dormancy and make a subheading or consider removing this part from the manuscript. It seems to me that the authors compares too many parameters (mesophilic and hyperthermophilic) as well as storage conditions 14 weeks and >1year, -80 and 4°C and with and without glycerol (the latter also including a extra treatment step of centrifugation that could also have an effect) to really make the conclusion concerning dormancy on lines 238-239.
12. Lines 253-255 Consider moving these lines to the introduction. See comment above
13. Line 282: consider adding a new subheading here to improve the readability of the manuscript.
14. Line 298: "*M. kandleri* should possess an S-layer,..." This statement is not clear. Are you refereeing to another study or is it your own observation. This needs to be better explained. Especially as the S-layer is lifted as a central result for the study in both abstract and introduction.
15. Line 358-359 The result could also be explained by sensitivity and therefore improving medium composition would not necessarily improve productivity as indicated in the sentence.
16. Line 401 Consider changing "have been" to "was" to clarify that the finding derived from this study on not refereeing to previous work. This is overall an issue with the text in general that should be rewritten keeping this aspect (weather the finding came from this study or it is referring to previous work) in mind.
17. Lines 401-407 Very long and difficult sentence that should be improved.
18. Line 418. Same comment as above consider replacing "should" with "are known to be" or similar to clarify that you refer to previous work.

19. Lines 510-513 The cooling and reheating of the cultures from thermophilic and hyperthermophilic conditions should affect the growth profile. The consequence of the cultivation should be included in the discussion of the results.
20. Line 552 Why did you use this equation has it previously been described. Please add reference.
21. Lines 604- 685 please provide the media composition in the supplement material and only add the essential information of the main differences between the composition in the main text.
22. Line 703 the accuracy of the HFG should be described. Did you do any calibration or other check of the accuracy of the machine this information would add to the quality of the manuscript. Also, pH was discussed in many places as an important factor correlated to pressure. Did you measure pH before or during cultivations?

Line 78. Misspelling, should be “steel”.

Line 81. suggesting add a reference at the end of the sentence.

Line 284. Please move the “surface protein layers (S-layers)” to an earlier place when it shows the first time.

Fig1. The resolution now is not high enough for reading the methanogens on the right side. Please provide a higher resolution figure.

Fig S2. Just curious, have you try mapping the name of methanogens on this figure, Is there any pattern showed according to phylogenetic relation?

Line 222. Similar to above comments, suggest adding methanogen names, maybe use a higher ranking level to reduces the possible name overlapping.

Line 265. Do you mean Fig.2a?

Line 266. Maybe also except for *Methanobacterium palustre* F and *methanocaldococcus vulcanius* M7 ?

Line 307. There is some discussion below Fig S7 in the supplementary. Consider adding it back to the main context.

Line 325. For the term “liquid-limitations,” do you mean the speed for H₂ and CO₂ diffuse into liquid is slower than the methane consumption speed? Maybe good to specify the definition somewhere.

Line 329. Considering list which are these methanogens.

Line 332. Repeat sentence, please remove.

Line 338. This is description for Fig2? Please revise.

Line 341. Methanogens font at this level should be italic.

Line 351. In Fig9S, some description of what is R1, R2, R3 and R4 need to be added in figure description.

Line 386. Suggesting give a subheading for each section just like the results part.

Line 434. There is an unrecognized symbol in FeS, please revise.

Line 489. Misspelling. Should be “through.”

Line 496. Misspelling. Should be “psychrophiles.”

Line 497. Misspelling. Should be "Psychrotolerant"

Line 518. Same as comments for Line 341.

Line 705. Misspelling. Should be "conversion".

Line 723. Misspelling. Should be "repressurized".

Referee expertise:

Referee #1: Biotechnology, microbiology, and renewable energy

Referee #2: Biotechnology & bioenergy

Referee #3: Bioinformatics, microbiology

Reviewers' comments:

Reviewer #1 (Remarks to the Author):	
General comment	Reply
The manuscript presents very interesting new results. Below specific comments which should be considered.	Dear reviewer 1, Thank you very much for your great input to improve our manuscript!
Specific Comments	Reply
1. The EU policy is to produce hydrogen as the fuel of the future due to the lack of CO ₂ emissions. In contrast to biomethane, hydrogen is emission-free. The use of hydrogen from RES for CO ₂ sequestration will seem pointless. It seems more reasonable to use hydrogen for temporary transformation into methane for its storage and transmission. Subsequently, its conversion to hydrogen would be analogous to the regasification of liquid methane to gas. One statement on page 3; line 63 - does not reflect the essence of the problem.	Thank you for the comment. We agree to your comment. indeed, hydrogen would be emission-free, and it is a renewable fuel that can be used, if it is not produced as grey or blue hydrogen. We re-wrote the introduction to be focussed on biomethanation using pure cultures, but we also highlighted mixed-culture applications.
2. Page 3; lines 72-74 - there is no literature reference to the information for the statement "This metabolic capability is essential when it comes to organic matter degradation in anoxic environments with low sulphate, nitrate, and Mn (IV) or Fe (III) concentrations." Some publications indicated an increase in methane production with the addition of iron filings.	Thank you for your comment! Yes, we are aware of the importance of trace elements and their influence of the performance of methanogenic archaea. Since certain trace elements are used in essential cofactor during methanogenesis. Although a too high concentration of theses metals could also result in the inhibition of methanogenesis.
3. Page 5; lines 122-123 - the variables lack the control of dissolved CO ₂ and H ₂ concentration for the mentioned control of homogeneous or heterogeneous growth. With the use of such a variety of pressurized media, a wide variety of chemical reactions could occur.	This is a good point. To investigate growth and CH ₄ productivity of each methanogen on the tested media, every closed batch cultivation was performed using three biological replicates (in some cases, 2 biological replicates) plus one negative control, as stated in material and methods. Therefore, we can conclude that the shown data has a profound validity. Of course, we also performed negative/zero control experiments.
4. Page 6; line 137 - subscript included in part of the sentence.	Every word in this sentence that is written with subscripted letters, should be written like that. Thus, no correction was needed.
5. Page 9; Fig.1 - illegible figure, I propose to break the figure to increase its legibility.	Thank you very much for your comment. We modified Fig. 1 by removing Fig. 1b, which was

	then included as a new the Supplementary material figure. We also uploaded the figures as individual images that you will be able to look at the Figure as high-resolution image.
6. Materials and methods mention the use of an H₂: CO₂ 4: 1 ratio. I did not find any information as to why this ratio was assumed. Bioresource Technology publication 233 (2017) 256–263 shows that only a higher ratio of 6: 1 to 10: 1 gives the best results for CO₂ conversion and CH₄ production. With such parameters, such rapid drops in pH would not have occurred, so that other species of methanogens could show a better result.	Thank you for your comment, it is really appreciated! We use a H₂/CO₂ ratio of 4 to 1 to be able to achieve a full conversion of substrate gases to product gas (please refer to Rittmann, S., Seifert, A., Herwig, C., 2015. Essential prerequisites for successful bioprocess development of biological CH₄ production from CO₂ and H₂. Crit. Rev. Biotechnol. 35, 141–151. https://doi.org/10.3109/07388551.2013.820685 and Seifert, A.H., Rittmann, S., Herwig, C., 2014. Analysis of process related factors to increase volumetric productivity and quality of biomethane with Methanothermobacter marburgensis. Applied Energy 132, 155–162. https://doi.org/10.1016/j.apenergy.2014.07.002). All other ratios will not result in full gas conversion. The mentioned publication gives important insights into the operational possibilities of in-situ biomethanation. Although, in this study we investigated pure culture methanogens for future applications in an ex-situ biomethanation process. We really like the idea of the concept to increase the H₂ ratio and thereby enhance the CH₄ productivity, and we already performed such experiments ourselves (Bernacchi, S., Rittmann, S., H. Seifert, A., Krajete, A., Herwig, C., 2014. Experimental methods for screening parameters influencing the growth to product yield (Y(x/CH₄)) of a biological methane production (BMP) process performed with Methanothermobacter marburgensis. AIMS Bioengineering 1, 72–86. https://doi.org/10.3934/bioeng.2014.2.72.) and analysed all data on pure culture biomethanation (Rittmann, S.K.-M.R., Seifert, A.H., Bernacchi, S., 2018. Kinetics, multivariate statistical modelling, and physiology of CO₂-based biological methane production. Applied Energy 216, 751–760. https://doi.org/10.1016/j.apenergy.2018.01.075)
7. In the supplement S1 to FIG1 with the HTS screening experiment to select medium, the information about temperature is missing.	All cultivation temperatures of the investigated methanogens are mentioned in materials and methods, as well as in Figure captions.
8. Authors should include supplement the protocols for testing the storage capacity of methanogen strains for continuation of	Thank you for your comment, we moved the paragraph on dormancy from the main text to the Supplementary material. This is in detail

research in the event of infection or contamination of experiments - page 10 verse 236-237.	described in the respective section in the Materials and Methods part.
9. Authors should include/crlarify supplement containing bioinformatic analysis protocols and proteomics demonstrating correlation of the presence of S surface proteins in the outer membrane with the increased ability of methanogens possessing them to a greater ability to reduce CO₂. Same Supplements regarding the demonstrated relationship between the specific set of amino acids Tyrα444 Glyα445 Tyrα446 in the construction of methyl-coenzyme M reductase (αMcr) proteins that are key for methanogens and increased conversion efficiency for archaea possessing it. Sets for all methanogens.	Thank you for your comment. The S-layer investigation was a search for S-layer using the UniProt platform, as described in detail in material and methods. The amino acid sequences that were used for the analysis of the alpha subunit of the Mcr are shown in Supplementary Material 2 (Excel file). The focus of the amino acid study also lied on the 80 investigated methanogens. We also added other representatives, to strengthen our findings. We aimed to find specific pattern that could underline the higher CH₄ productivity that we found during practical investigations with a bioinformatic analysis. We analysed representatives from 32 different genera. Thus, we give already a good overview about the presence or exchange of Tyr^{α444}. Representatives from different genera:  Methanomassilicoccus Methanococcus Methanotorris Methanocaldococcus Methanobacterium Methanothermobacter Methanothermococcus Methanothermolithococcus Methanobrevibacter Methanopyrus Methanothermus Methanofervidococcus Methanotrix Methanosaeta Methanohalophilus Methanosalsum Methanomethylovorans Methanococcides Methanolobus Methanimicrococcus Methanosarcina Methanocella Methanoregula Methanolinea Methanocorpusculum Methanospirillum Methanolancinia Methanoregula Methanogenium Methanospherula Methanoculleus Methanofollis

10. There are some minor typing errors.

Thank you, those have been corrected.

Reviewer #2 (Remarks to the Author):	
General comment	Reply
The manuscript assesses high pressure methane-generating cell factories for H₂/CO₂ conversion using single cell cultures. 80 different species were evaluated for their performance ranging from psychrophilic to hyper thermophilic range and provides valuable insights on the effect of medium composition and elevated pressures as well as phylogenetic characteristics of high performing cell-factories. The manuscript provides comprehensive data and many insights that are expected to be valuable for the continuation technical development in this field as well as contributing to the fundamental knowledge of these methanogenic archaea. Overall, the manuscript provides relevant and novel information that is expected to gain interest from both the academic community as well as being industrially relevant.	Dear reviewer #2! Thank you very much for your time and help to improve our manuscript.
However, the results and discussion are difficult to follow and the English needs to be improved.	Thank you for your comment, it is very appreciated. We rewrote the introduction and shortened the results section by moving information to the Supplementary material section. We hope having improved the legibility by focussing on the main message of the manuscript. Moreover, English proof reading was performed by a native speaker.
It is also difficult in places to distinguish between what is result or conclusions originating from the current work and what is referring to previous work.	Thank you for your comment! Please see comment above.
In general, the introduction needs to be more focused and relevant for the current work and the results section needs to be dived in more sections with subheadings and better structured to facilitate the readability.	Thank you for your suggestions! The introduction is specifically focused on the current work. We have completely rewritten the introduction. The result section was divided into several subsections by adding new headers, and we moved two parts from the results section to Supplementary material.
Moreover, figures are difficult to read when printed out. Consider using bigger font and distinguishable bars.	Thank you for the comment. Indeed, this was a problem when uploading a word document in which the figures were embedded. Actually, the figures have had a very high resolution – and still have. We uploaded the figures as separate files and hope this has now solved a part of the legibility issues. We hope having improved the legibility of the figures by moving one heatmap of Fig. 1 to the Supplementary material. We introduced grey and black grid lines to Fig.1 and separated the figure into temperature groups. We separated the bar charts in Fig. 2 and Fig. 3

	and increase the font size of these figures.
Enough information is given in the material and method section to reproduce the work. However, some issues related to the methods should be included in the discussion of the result as they may be contributing factors. Please refer to the specific comments below.	Thank you for this comment, we addressed the specific comments.
Please also consider mentioning the media optimization and the effect of the specific nutrient additions in the abstract since it constitutes a large part of the result content and elaborate more on the understanding these results with more references to previous work in both the introduction and discussion sections.	Thank you. Unfortunately, the abstract is only allowed to contain 150 words. Please accept our apologies that we even had to shorten the abstract.
Specific comments	Reply
1. Line 35: CH ₄ instead of CH4.	Thank you for your comment, it has been corrected.
2. Line 35: In the abstract, the S-layer and MCR α are the only specific features mentioned from the bioinformatic study. Why is not the role of selenium in the hydrogenase enzyme mentioned?	Thank you very much for your comment. We do not mention the selenium because this is a discussion point and not a finding of our manuscript.
3. Lines 49-61: This text does not introduce the current work and is not contributing to the understanding of the current work. Removing this would reduce the length of an already extensive introduction section and allow for the importance and relevance of the current work to be introduced to the reader.	Thank you for your suggestion. The mentioned lines and citations (see below) were deleted. "Thermogenic CH ₄ emission sources are terrestrial and marine seeps, volcanic activity, and fossil fuel exploitation. Pyrogenic CH ₄ originates from wildfires (soil carbon), incomplete combustion, and from leaking biomethane and natural gas ³ . CH ₄ is degraded to carbon dioxide (CO ₂) in the atmosphere and recycled or deposited in natural sinks ^{4,5} . The troposphere is the largest CH ₄ sink. There, hydroxyl radicals degrade 90% of emitted CH ₄ over some intermediate steps to CO ₂ . Furthermore, CH ₄ is degraded by methanotrophic bacteria in aerated soils (4%), through chemical reactions with chlorine and atomic oxygen radicals in the stratosphere (3%), and by destruction through chlorine radicals from sea salt in the marine boundary layer to the atmosphere (3%) ³ . Although CH ₄ has a short lifetime of 6.3 to 12.5 years in the atmosphere ^{4,5} , it has a 28 times higher global warming potential compared to CO ₂ ⁶ . Latest observations indicate a growing imbalance between CH ₄ emissions and sink capacity. After a stabilizing period, atmospheric CH ₄ began to rise in 2007, accelerated between 2014 to 2017

	at annual growth rates (10 ppb year ⁻¹) 7. It is estimated that anthropogenic fossil CH ₄ emissions account for about 30% of the global CH ₄ 8.”
4. Lines 79-84: In an AD process an undefined multi-strain consortium is more flexible and robust. In this text you just mentioned to positive sides for pure cultures. To be more convincing in your pure-culture strategy, it would be good to mention the positive argument of a diverse culture for example reduced risk of contamination. This would be interesting to put in relation to why for example growth and MES are relevant factors to study.	Thank you for your comment! Your comment was addressed. However, we want to emphasise that there are actually almost no contamination issues when it comes to growing methanogenic archaea in pure culture on defined media on H ₂ /CO ₂ .
5. Lines 84-91: Similarly, to above you only mentioned the negative issues with Sabatier. By variegate the information and also mentioned for example lower rate of reaction of the biological system the essence of the current study would be better introduced.	Thank you for your comment. We also mentioned some positive aspects of the Sabatier reaction.
6. Lines 97-98: Please provide reference. Moreover, here is the only place in the introduction where you mentioned high pressure. However, since this is a very important part of this study the reader needs more references to previous work, i.e what is previously known about pressure tolerance and H ₂ availability and how the current study further increases the knowledge on this aspect. Similarly, the effect of nutrient addition and media composition should be mentioned. What is the current knowledge of these additives and the effect of rich v.s. mineral medium?	These lines were referenced and we focused more on pressure in the introduction section.
7. Line 122: You are using a high H ₂ /CO ₂ ratio. Since this is a fundamental study the high ratio is accepted. However, since this ratio might not be industrially possible due to the expense of hydrogen, this issue should be better discussed in possible the introduction but clearly discussion and conclusion sections.	Thank you for the comment. We use a H ₂ /CO ₂ ratio of 4:1 because we want to achieve a full conversion of educt gas to product gas and the only possibility is to utilize a 4:1 H ₂ :CO ₂ ratio. Please refer to: Rittmann, S., Seifert, A., Herwig, C., 2015. Essential prerequisites for successful bioprocess development of biological CH ₄ production from CO ₂ and H ₂ . Crit. Rev. Biotechnol. 35, 141–151. https://doi.org/10.3109/07388551.2013.820685 Seifert, A.H., Rittmann, S., Herwig, C., 2014. Analysis of process related factors to increase volumetric productivity and quality of biomethane with Methanothermobacter marburgensis. Applied Energy 132, 155–162. https://doi.org/10.1016/j.apenergy.2014.07.002
8. Line 125 According to the equation 1 Line	Thank you for your comment, you are right it is

552 the units for the biomass increase rate is incorrect it should be unitless. Please check and update in all places.	unitless.
9. Line 115-248 this section is very long and difficult to follow. Please consider dividing it in more sections with subtitles to improve readability.	Thank you for your comment. This was done.
10. Line 132. Please mention MER also for thermophiles as you include this result for all other groups. Fig 1 contains very relevant and important information but is very difficult to read. Please consider diving it in more plots and increase the font size.	The MER for thermophiles is included. Fig. 1 includes now OD_{max} and MER_{max}, the $turnover_{max}$ heatmap was moved to the supplementary material. For better readability of all heatmaps, light grey gridlines were added. Additionally, the results of psychrophiles, mesophiles, thermophiles and hyperthermophiles were separated by black lines.
11. Line 230-239 please group the results on effect of dormancy and make a subheading or consider removing this part from the manuscript. It seems to me that the authors compares too many parameters (mesophilic and hyperthermophilic) as well as storage conditions 14 weeks and >1year, -80 and 4°C and with and without glycerol (the latter also including an extra treatment step of centrifugation that could also have an effect) to really make the conclusion concerning dormancy on lines 238-239.	Thank you for your comment! This paragraph was moved to the supplementary material, since it is an important information, especially when considering methanogens as biocatalysts for industrial application. The reactivation ability after dormancy is also essential, when investigating methanogens in the laboratory. Therefore, we would rather keep this subsection, at least in the supplementary material.
12. Lines 253-255 Consider moving these lines to the introduction. See comment above	In these lines results are described, thus they should be kept in the result section.
13. Line 282: consider adding a new subheading here to improve the readability of the manuscript.	Thank you for your comment! Several new subheadings have been added to improve the readability of the manuscript.
14. Line 298: "M. kandleri should possess an S-layer,..." This statement is not clear. Are you refereeing to another study or is it your own observation. This needs to be better explained. Especially as the S-layer is lifted as a central result for the study in both abstract and introduction.	The referring citation has been added and it is now also mentioned in the text that we are referring to another publication. In this study, we could not find any S-layer specific motifs or domains as it was the case for other methanogens that are known to be covered with a S-layer.
15. Line 358-359 The result could also be explained by sensitivity and therefore improving medium composition would not necessarily improve productivity as indicated in the sentence.	M. marburgensis is probably more sensitive towards pressure (50 bar) as Methanocaldococcus spp., which is also indicated by the pressure curves in Fig. S9. In the first repetitive closed batch run (RCB1), M. marburgensis showed an extremely long lag-

	phase of nearly 200 hours, whereas Methanocaldococcus spp. showed almost no lag-phase. In the second run RCB2 Methanocaldococcus spp. showed much flatter pressure curves as in the previous run (RCB1), thus media adaption toward high-pressure could results in a consistent or even an increased CH₄ productivity.
16. Line 401 Consider changing "have been" to "was" to clarify that the finding derived from this study on not refereeing to previous work. This is overall an issue with the text in general that should be rewritten keeping this aspect (weather the finding came from this study or it is referring to previous work) in mind.	In the mentioned line "have been" was changed to "were". Also, special emphasis was put on the "has been/ have been" and "was/ were" issue, when reading the text, to avoid confusions. Previous work was always cited.
17. Lines 401-407 Very long and difficult sentence that should be improved.	Thank you, we tried to shorten long sentences throughout the manuscript.
18. Line 418. Same comment as above consider replacing "should" with "are known to be" or similar to clarify that you refer to previous work.	To clarify that we refer to a previous work in the mentioned line "should" was changed to " is known to be " .
19. Lines 510-513 The cooling and reheating of the cultures from thermophilic and hyperthermophilic conditions should affect the growth profile. The consequence of the cultivation should be included in the discussion of the results.	This is a very good point and we are aware of this issue and the resulting consequences. Therefore, our analysis only considered the kmin for MERmax calculations and not the global time for full conversion to account for this phenomenon. The heating and cooling times were not considered in our analyses.
20. Line 552 Why did you use this equation has it previously been described. Please add reference.	This equation has not been previously described. We introduced this equation in this manuscript.
21. Lines 604- 685 please provide the media composition in the supplement material and only add the essential information of the main differences between the composition in the main text.	The media compositions were moved to supplementary material 1. Additionally, essential information of the major differences between media compositions were mentioned in the main text. We also shortened the respective results sections.
22. Line 703 the accuracy of the HFG should be described. Did you do any calibration or other check of the accuracy of the machine this information would add to the quality of the manuscript. Also, pH was discussed in many places as an important factor correlated to pressure. Did you measure pH before or during cultivations?	We assume that the reviewer meant the SBRS and not the HFG. About the accuracy of the SBRS: the SBRS consist out of four identical bioreactors, which are equipped with an online pressure sensor. Besides, that a manometer is installed. The pressure sensor and the manometer were calibrated, before installation. The accuracy of the heating jacket was also tested before the experiments were performed. The comparison of the digital pressure sensors between the bioreactors and the manometer provides proof that pressure is accurately measured. The fact that each experiment is also performed in quadruplicates, strengthens the

validity of the results. Some high-pressure experiments could not be investigated with all four bioreactors due to technical or biological failure. Since this was a screening approach, high performance methanogens were further investigated and their growth and CH₄ production kinetics were then analysed in detail.

Indeed, pH was discussed in many places, due to its importance for the performance of methanogenic pure cultures in the context of hyperbaric H₂/CO₂ conversion. Under hyperbaric cultivation conditions, pH was not measured, as the SBRS is not equipped with a pH probe. Therefore, pH control or possible feeding was not possible with this bioreactor system. Anyway, it is a closed batch system. *Taubner et al. 2018* could show that *M. okinawensis* was able to grow at hyperbaric conditions of 50 bar, although p_{CO2} analyses revealed an extremely low pH of 3.5 under these conditions.

Reviewer #3 (Remarks to the Author):	
General comments	Reply
The experiment of this manuscript is well designed and presented. The results are also adequately discussed. However, the reviewer suggests improving the English language and correcting several minor mistakes before publication.	Thank you very much for performing the review! We are grateful to your comments! The mistakes have been corrected. English proof reading by a native speaker has been performed.

Additional changes	
Introduction	Due to the restructuring and refinement of the introduction the order of the citations changed, and additional citations were added.
Fig. S8	The 10 bar pressure curves of of M. jannaschii (RCB1 and RCB2) changed. The first version of Fig. S8 did not include the latest pressure curves of M. jannaschii .
Line 78. Misspelling, should be "steel".	Thank you that was corrected.
Line 81. suggesting add a reference at the end of the sentence.	Thank you. During the revisions we added several new references to statement sentences.
Line 284. Please move the "surface protein layers (S-layers)" to an earlier place when it shows the first time.	Thank you for the comment. We abbreviated S-Layer earlier in the text, however not in the results section header and we used the abbreviation S-Layer in the abstract due to space restrictions.
Fig1. The resolution now is not high enough for reading the methanogens on the right side. Please provide a higher resolution figure.	Thank you for the comment. We agree that the resolution was not high enough in the submitted word document. Therefore, we uploaded the figures individually as high-resolution images. Now every font should be legible upon magnification.
Fig S2. Just curious, have you try mapping the name of methanogens on this figure, Is there any pattern showed according to phylogenetic relation?	Thank you for the comment. We performed multivariate statistical analysis to analyse if a pattern would emerge due to phylogenetic constraints. This was partially possible and the results are presented in Supplementary material files and partially in the manuscript.
Line 222. Similar to above comments, suggest adding methanogen names, maybe use a higher ranking level to reduces the possible name overlapping.	Thank you. We increased the figure quality and we hope that our high-resolution images made the legibility of the strain names legible
Line 265. Do you mean Fig.2a?	We meant Fig. 2.
Line 266. Maybe also except for Methanobacterium palustre F and methanocaldococcus vulcanius M7?	We are not absolutely certain about the meaning of your comment. However, as the figures were now uploaded as high-resolution images, we hope to have increased their legibility.
Line 307. There is some discussion below Fig S7	Thank you very much for the comment. As we

in the supplementary. Consider adding it back to the main context.	have to adhere to editorial size restrictions, we had to move some paragraphs from the main text to the Supplementary material. Please allow us that some content is only presented in these documents.
Line 325. For the term “liquid-limitations,” do you mean the speed for H ₂ and CO ₂ diffuse into liquid is slower than the methane consumption speed? Maybe good to specify the definition somewhere.	Thank you for the comment. Liquid limitation refers to nongaseous nutrients.
Line 329. Considering list which are these methanogens.	Thank you. However, line 329 referred to a sentence in the figure captions in V1 of this manuscript in which no methanogens were mentioned.
Line 332. Repeat sentence, please remove.	Thank you, done.
Line 338. This is description for Fig2? Please revise.	Thank you very much or pointing this out. This was changed.
Line 341. Methanogens font at this level should be italic.	Thank you this was changed.
Line 351. In Fig9S, some description of what is R1, R2, R3 and R4 need to be added in figure description.	Thank you for the comment. This was added to the figure header.
Line 386. Suggesting give a subheading for each section just like the results part.	Thank you. As we are on the boundary to the maximum word limit we kindly ask not to add subheadings within the discussion section.
Line 434. There is an unrecognized symbol in FeS, please revise.	Corrected
Line 489. Misspelling. Should be “through.”	Corrected
Line 496. Misspelling. Should be “psychrophiles.”	Corrected
Line 497. Misspelling. Should be “Psychrotolerant”	Corrected
Line 518. Same as comments for Line 341.	Corrected
Line 705. Misspelling. Should be “conversion”.	Corrected
Line 723. Misspelling. Should be “repressurized”.	Corrected

REVIEWERS' COMMENTS:

Reviewer #1 (Remarks to the Author):

In my opinion, the manuscript was appropriately revised and is now suitable for publication.

Reviewer #2 (Remarks to the Author):

I agree with the update of the manuscript. I do not have any additional comments